# On Theoretical Identifiability of Binary Latent Causal Graphical Models

**Seunghyun Lee**  *sl4963@columbia.edu*
*Department of Statistics*
*Columbia University*

**Yuqi Gu**  *yuqi.gu@columbia.edu*
*Department of Statistics*
*Columbia University*

**Reviewed on OpenReview:** *https://openreview.net/forum?id=KiiSlAsLuN*

## Abstract

This paper considers a challenging problem of identifying a causal graphical model under the presence of latent variables. While various identifiability conditions have been proposed in the literature, they often require multiple pure children per latent variable or restrictions on the latent causal graph. Furthermore, it is common for all observed variables to exhibit the same modality. Consequently, the existing identifiability conditions are often too stringent for complex real-world data. We consider a general nonparametric measurement model with arbitrary observed variable types and binary latent variables, and propose a *double triangular* graphical condition that guarantees identifiability of the entire causal graphical model. The proposed condition significantly relaxes the popular pure children condition. We also establish necessary conditions for identifiability and provide valuable insights into fundamental limits of identifiability. Simulation studies verify that latent structures satisfying our conditions can be accurately estimated from data. We also illustrate the practicality of our conditions with a real data example.

## 1 Introduction

Discovering causal relationships under the presence of latent variables is a fundamental challenge in observational studies such as psychology and education, as well as modern machine learning and generative modeling (Bollen, 2002; Pearl, 2014). Latent variables are especially valuable in these domains as they offer a low-dimensional, interpretable representation that captures substantively meaningful concepts and effectively summarizes high-dimensional data. For instance, in educational assessments designed to evaluate students' understanding of key concepts, only their responses are observable, whereas the underlying conceptual knowledge is latent. Often, these latent variables have interpretable causal relationships among themselves. Consequently, an important problem is to identify this latent graph structure using only the noisy, observed measurements.

This problem – also commonly known as *identifiability of latent causal structures* – has been studied from numerous perspectives. Traditional studies have largely focused on models with continuous variables with linear or additive structural equations (Silva et al., 2006; Anandkumar et al., 2013; Huang et al., 2022; Xie et al., 2022; 2024; Montagna et al., 2023; 2025; Moran & Aragam, 2026), and established various conditions that guarantee model identifiability. Despite the attractive parsimony, this modeling assumption can be unrealistic in practice, as the latent concepts can be discrete and the relationship between the latent and the observed may be non-additive. For example, in educational assessments, it is common to consider binary latent variables to model cognitive abilities, whose values correspond to mastery or deficiency of a certain skill, say addition or multiplication (Junker & Sijtsma, 2001; von Davier, 2008). These models are often

highly non-linear, motivated by the fact that solving a question correctly requires the interaction of multiple latent skills.

Consequently, causal discovery with *discrete* latent variables has received growing attention in recent years. As causal graphical models with latent variables are non-identifiable without further assumptions (Spirtes et al., 2001), existing works have focused on proposing sufficient conditions for identifiability. In particular, common identifiability conditions for discrete latent variable models include (a) structural constraints on the latent graph, such as trees (Pearl & Tarsi, 1986; Song et al., 2014; Wang et al., 2017) or graphs with no triangles alongside atomic cover assumptions (Kong et al., 2024), (b) requiring two or three pure children (observed variables with exactly one latent parent) per latent variable (Gu & Dunson, 2023; Chen et al., 2024; 2025; Lee & Gu, 2026), or (c) assuming a mixture oracle that recovers the order of all marginal modes (Kivva et al., 2021)[1]. However, these existing results do not guarantee identifiability for complex causal structures where both the latent graph and the latent-to-observed graph are allowed to have many edges. For instance, only Kivva et al. (2021) can identify the graphical structure in Figure 1, by relying on the strong assumption of a mixture oracle.

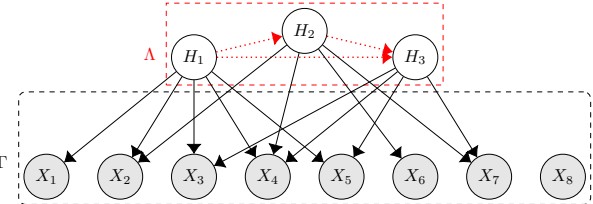

Figure 1: Example of an identifiable causal graphical structure. Here, $H = \{H_1, H_2, H_3\}$ denotes the unobserved hidden/latent variables, and $X = \{X_1, \ldots, X_8\}$ denotes the observed responses. The edges of the DAG are partitioned into the latent graph $\Lambda$ (indicated by the dotted red arrows) and the latent-to-observed bipartite graph $\Gamma$ (indicated by the solid black arrows).

This work proposes a new and significantly weaker set of conditions under which all components of the causal graphical model (both the graphs and conditional probabilities) are identifiable. We work under a setting where the latent variables are binary but the observed responses can be arbitrary and have flexible non-linear distributions. Binary latent variables are common in many applications, and can model a students' mastery/non-mastery of a skill or a patients' presence/absence of a disease. In particular, our proposed identifiability result shows that complex causal structures such as that in Figure 1 can be identified.

**Organization and main contributions** This paper is organized as follows. Section 2 formalizes the model setup and assumptions. Section 3 provides our main theoretical contributions regarding identifiability, which we further elaborate below.

(a) We propose *sufficient conditions for identifiability* without any parametric or latent structural requirements. Our main condition is the *double-triangular* graphical condition, which requires two distinct triangular structures in the latent-to-observed graph $\Gamma$ (see Theorems 1 and 2). These conditions are easily checkable and significantly weaker than those in the existing literature. In particular, we make no assumptions on the latent causal structure, do not require pure children or parametric forms, and allow arbitrary response types for the observed variables. Thus, our framework is broadly applicable to a wide range of real-world problems.

(b) We also propose *necessary conditions for identifiability*. In Theorems 3 and 4, we show that each latent variable must have at least three observed children, and the latent-to-observed graph must have non-nested columns (the so-called subset condition, see Definition 4). While these necessary conditions do not exactly match our sufficient conditions, they help depict the fundamental limits of identifying causal models with binary latent variables.

---

[1]While there exist identifiability results for *linear* models with *continuous* latents that do not fall into the three categories (a)–(c) (e.g. see Anandkumar et al. (2013); Adams et al. (2021); Li et al. (2025)), they do not extend to *nonlinear* models with *discrete* latents.

In Section 4, we conduct experiments on both simulated and real data to demonstrate the implications and practical usefulness of our identifiability theory. Finally, Section 5 discusses directions for future work.

## 1.1 Other related works

**Structure learning with latent variables** Learning graphical structures in the presence of latent variables has been widely studied, and there exist many works that learn (a) the latent structure up to a certain class of ambiguities (e.g. ancestral graphs in Richardson & Spirtes (2002) and Ali et al. (2009) or atomic covers in Xie et al. (2022)) or (b) the graphical structure among observed variables (Evans, 2018; Frot et al., 2019; Gordon et al., 2023). In particular, Evans (2018) considered a flipped setting (compared to ours) with discrete observations and arbitrary latent variables. However, our problem of identifying the causal structure *over all variables*, including those related to latent variables, is distinct from these line of works. Mainly, we work under the so-called "measurement model" (see Assumption 1) to resolve such ambiguities and identify the complex causal relations among the latent variables.

**Identifiability of hierarchical latent structures** Recently, identifiability issues have been explored for hierarchical latent structure models that allow latent variables without any observed children (Anandkumar et al., 2013; Xie et al., 2022; Kong et al., 2024; Xie et al., 2024; Lee & Gu, 2026; Dong et al., 2023). Under our general setup without any structural/parametric assumptions, it is impossible to recover hierarchical structures since such latent variables not directly measured by observed variables can be marginalized out without changing the likelihood, and we do not consider this direction.

| Paper \ Criteria | Know $K$ | Assume pure children | Response type restrictions |
|---|---|---|---|
| Kivva et al. (2021) | X | X | Continuous |
| Chen et al. (2024) | X | O | Discrete |
| Chen et al. (2025) | X | O | X |
| Lee & Gu (2026) | O | $\triangle^2$ | X |
| Ours | X | X | X |

Table 1: Comparison of identifiability results for directly related works with discrete latent variables. **Legend:** O = Yes/required, X = no/not-required, $\triangle$ = partial.

Directly related works that consider discrete latent variables (Kivva et al., 2021; Chen et al., 2024; 2025; Lee & Gu, 2026) are discussed in detail in Appendix A. In Table 1, we summarize the setup and assumptions of these works and compare with the current paper.

# 2 Background

## 2.1 Model setup and assumptions

We start by defining notations for the considered causal graphical models. Let $G = (V, E)$ be a directed acyclic graph (DAG), where the vertex set is partitioned as $V = (X, H)$, with observed variables $X = (X_1, \ldots, X_J) \in \prod_{j=1}^{J} \mathcal{X}_j$ and binary hidden/latent variables $H = (H_1, \ldots, H_K) \in \{0, 1\}^K$. As mentioned in the introduction, positing binary latent variables are common for modeling discrete concepts, for example to model mastery/deficiency of cognitive skills in educational testing data. This also turns out to be crucial for our technical arguments as well as for allowing minimal restrictions on observed sample space $\mathcal{X}_j$s, as further elaborated in Appendix G. The observed variables are not necessarily discrete, and can be *arbitrary* types of responses, as long as each sample space $\mathcal{X}_j$ is a nondegenerate separable metric space. Throughout the paper, we will work under this setting, which can allow flexible modeling of multi-modal data that includes both discrete and continuous responses.

---

[2]While the strict identifiability therein requires pure children, this is relaxed for the generic result.

We separate our global model assumptions into those on the graph structure $G$ and that on the probability distribution on the vertices $V$. First, regarding the graph $G$, we work under the following *measurement model assumption* (Silva & Scheines, 2004; Silva et al., 2006; Kivva et al., 2021).

**Assumption 1** (Measurement model). *Assume that there are no edges (a) between the observed variables $X$ and (b) from observed to hidden variables.*

Assumption 1 states that the observations are noisy measurements of the latent. This assumption is natural in domains including psychometrics and medicine, where observable test answers (or symptoms) do not cause one another, but are instead common effects of unobservable underlying skills (or diseases). Then, we decompose the edge set as $E = \Lambda \cup \Gamma$, where $\Lambda$ collects the edges among the hidden variables $H$ (i.e. the latent causal graph), and $\Gamma$ collects the edges from one hidden variable to one observed variable (i.e. the latent-to-observed bipartite graph). With a slight abuse of notation, also let $\Gamma$ denote the $J \times K$ bipartite graph adjacency matrix where the $(j, k)$-th entry $\gamma_{j,k}$ is an indicator of whether there is an edge from $H_k$ to $X_j$. See Figure 1 for a visualization. Note that an observed variable can have *all* latent variables as parents ($X_4$) or *no* parent ($X_8$).

Next, we discuss assumptions on the probability distribution $\mathbb{P}(V) = \mathbb{P}(X, H)$. We assume that $\mathbb{P}(V)$ follows the causal Markov property under $G$ and is faithful, which are fundamental assumptions for learning graphical models (Spirtes et al., 2001).

**Assumption 2** (Basic graphical model assumptions). *Assume the causal Markov property*

$$\mathbb{P}(V) = \prod_{V_i \in V} \mathbb{P}(V_i \mid \mathsf{pa}(V_i))$$

$$= \prod_{k=1}^{K} \mathbb{P}(H_k \mid \mathsf{pa}(H_k)) \prod_{j=1}^{J} \mathbb{P}(X_j \mid \mathsf{pa}(X_j)).$$

*Also, assume that $\mathbb{P}(V)$ is faithful to $G$ in the sense that the conditional independence relationships embedded in the DAG $G$ is equal to that in $\mathbb{P}(V)$.*

We additionally impose the following nondegeneracy assumption.

**Assumption 3** (Nondegeneracy). *The probability distribution of $V = (X, H)$ satisfies:*

   *(a) All latent configurations are allowed, that is $\mathbb{P}(H = h) > 0$ for all $h \in \{0, 1\}^K$.*
   *(b) For each $j$, the conditional distributions of $X_j$ are distinct across latent configurations: $\mathbb{P}(X_j \in C_j \mid \mathsf{pa}(X_j) = h) \neq \mathbb{P}(X_j \in C_j \mid \mathsf{pa}(X_j) = h')$ for all $h \neq h' \in \{0,1\}^{|\mathsf{pa}(X_j)|}$ and $\phi \neq C_j \subsetneq \mathcal{X}_j$.*
   *(c) Each latent variable $H_k$ has at least one observed child. In other words, $\Gamma$ has no all-zero columns.*

We elaborate more on each condition. Here, Assumption 3(a) ensures full support over all latent configurations. Assumption 3(b) ensure nondegeneracy of all conditional distributions that correspond to each edge in the latent-to-observed graph $\Gamma$. Note that we only require the conditional distributions of $X_j$ to be distinct for each configuration of the latent parents, and do not assume full rankness of the matrix $\mathbb{P}(X_j \mid \mathsf{pa}(X_j))$. These two conditions are standard for identifying discrete latent causal models (e.g. see Assumption 2.4 in Kivva et al. (2021) or Condition 4.1 (i) in Chen et al. (2024)). We refer the interested reader to Appendix A in Kivva et al. (2021) for examples arguing the necessity of these conditions. In particular, in Appendix B of this paper, we provide an alternative example with binary responses that highlights the need for the second condition. Assumption 3(c) rules out unmeasured latent variables without any observed child. While this is often considered as part of the measurement model setting, we spell this out to clarify that no hierarchical (or multi-layer) latent structures are considered, as discussed in Section 1.1.

Based on Assumption 3, we formalize the class of models that we will consider as *binary latent causal models*.

**Definition 1** (BLCM). *A Binary Latent Causal Model (BLCM) on the graph $G = (V, E)$ is a probability distribution $\mathbb{P}(V)$ that satisfies Assumptions 1–3.*

**Remark 1** (Comparison with assumptions in the literature). *One distinction of this paper compared to other discrete latent causal models (Kivva et al., 2021; Chen et al., 2024; Kong et al., 2024) is that we assume*

*a priori that all latent variables are* binary. *While this is a stronger assumption, it is highly interpretable and also grants additional flexibility. To elaborate, the cited papers require additional conditions on top of Assumption 3, such as no twins and maximality, to deal with the merging and splitting of latent variables. We do not require such assumptions.*

*Also, note that we do not impose any structural assumptions on the latent graph* $\Lambda$; *it can be an arbitrary graph that ranges from a tree to a complete graph, and can even allow isolated latent variables (without any latent neighbor). This stands in contrast to many previous approaches that restrict the latent graph to be a tree (Pearl & Tarsi, 1986; Choi et al., 2011) or forbids triangles (Kong et al., 2024).*

## 2.2 Objective: Identifiability

Under the above modeling assumptions, our goal is to identify all model components of the BLCM from the observed distribution $\mathbb{P}(X)$:

1. The number of latent variables $K$,
2. The latent DAG $\Lambda$ and the latent-to-observed bipartite graph $\Gamma$,
3. The distribution of the latent variables $\mathbb{P}(H)$ and the conditional observed distributions $\mathbb{P}(X_j \mid H)$.

Here, it is important to clearly define the notion of "identifiability", as the exact definition varies by different models and papers. We wish to adopt the most stringent, statistical notion of parameter identifiability. In other words, by viewing the above model components as a parameter vector $\theta$, we say that an identical observed distribution $\mathbb{P}(X; \theta) = \mathbb{P}(X; \theta')$ implies identical parameters $\theta = \theta'$.

Unfortunately, identifiability in latent variable models is subject to various fundamental ambiguities, and the above definition does not directly apply. First, we consider identifiability up to latent variable *label permutation* (e.g., $H_1$ and $H_2$ can be re-labeled in Figure 1) and *sign-flipping* (e.g., the meaning of $H_1 = 0$ and $H_1 = 1$ can be permuted). These two ambiguities are fundamental under the presence of categorical latent variables (Lee & Gu, 2026). Additionally, in terms of identifying the graphical structure, we identify the latent graph $\Lambda$ up to its *Markov equivalence* class. It is well known that the edges $E = (\Lambda, \Gamma)$ are identifiable only up to Markov equivalence, even when the entire distribution $\mathbb{P}(X, H)$ is known (Pearl, 2014; Spirtes et al., 2001). Under the measurement model assumption described earlier (Assumption 1), the bipartite graph $\Gamma$ does not suffer from this problem and can be fully identified. A rigorous definition of these trivial ambiguities are provided in Appendix C.1.

Based on the above considerations, we formally define the notion of identifiability that will be used in the remainder of the paper.

**Definition 2** (Identifiability). *Consider a BLCM with distribution* $\mathbb{P}(V)$ *on* $G = (V, E)$, *with variables* $V = (X, H)$ *and edges* $E = \Lambda \cup \Gamma$. *We say that the* latent dimension $K$ *is identifiable if there is no alternate BLCM with distribution* $\widetilde{\mathbb{P}}(\widetilde{V})$ *on* $\widetilde{G} = (\widetilde{V}, \widetilde{E})$ *that satisfies the following:*

*(a)* $\widetilde{V} = (X, \widetilde{H})$ *for latent variables* $\widetilde{H} \in \{0, 1\}^{\widetilde{K}}$ *such that* $\widetilde{K} < K$.

*(b)* $\mathbb{P}(X) = \widetilde{\mathbb{P}}(X)$, *that is,* $\mathbb{P}$ *and* $\widetilde{\mathbb{P}}$ *define identical marginal distributions.*

*Next, given the latent dimension* $K$, *we say that the remaining* model components $(E, \mathbb{P})$ *are identifiable when there is no alternate BLCM with distribution* $\widetilde{\mathbb{P}}(\widetilde{V})$ *that satisfies the above condition (b) and the following conditions (a'), (c):*

*(a')* $\widetilde{V} = (X, \widetilde{H})$ *for latent variables* $\widetilde{H} \in \{0, 1\}^K$.

*(c)* $(E, \mathbb{P})$ *is distinct from* $(\widetilde{E}, \widetilde{\mathbb{P}})$ *up to label permutation, sign-flipping, and Markov equivalence (see Definition 5 for details).*

Definition 2 ensures that no alternate, nontrivial BLCM with $\widetilde{K} \leq K$ latent variables defines the same observed distribution $\mathbb{P}(X)$. The alternate latent dimension $\widetilde{K}$ is restricted to address a final source of trivial non-identifiability that arises from the existence of redundant latent variables (e.g. see Lemma 1 in Evans (2016), or the notion of minimality in Markham & Grosse-Wentrup (2020)). In other words, we can always

create redundant latent variables by splitting existing ones. This restriction can be avoided at the cost of imposing additional constraints on alternate BLCMs (see Remark 4 and Proposition 2).

## 2.3  Notations

For an integer $K$, write $[K] := \{1, \ldots, K\}$. Recall that $V = (X, H)$ collects all observed and hidden variables. For any variable $v \in V$, we use standard graphical model notations and let $\mathsf{pa}(v), \mathsf{ch}_X(v)$ denote the parent variables and observed children (in $X$) of $v$, respectively. For example, in the setting of Figure 1, we have $\mathsf{pa}(X_2) = \{H_1, H_2\}$ and $\mathsf{ch}_X(H_2) = \{X_2, X_4, X_6, X_7\}$. Let $H_{(-k)} := H \setminus \{H_k\}$ denote the set of all hidden variables except for $H_k$. Write $\mathbb{P}_{j,h} := \mathbb{P}(X_j \mid H = h)$ as the conditional probability for the observation $X_j$, and let $\pi_h := \mathbb{P}(H = h)$ be the probability of the latent configuration $h$. Under categorical observations with $|\mathcal{X}_j| < \infty$, for any $S \subseteq [J]$, let $\mathbb{P}(X_S \mid H)$ denote the conditional probability table whose rows and columns are indexed by $\prod_{j \in S} \mathcal{X}_j$ and $\{0,1\}^k$, respectively. Let $\pi = (\pi_h)_{h \in \{0,1\}^K}$ collect all mixture proportions, and let $\mathsf{diag}(\pi)$ be the $2^K \times 2^K$ diagonal matrix with entries of $\pi$ on the diagonal. For two binary vectors $h, h'$ of the same length, write $h \succeq h'$ when $h_k \geq h'_k$ for all $k$.

# 3  Main Identifiability Results

## 3.1  Sufficient Condition

We present our main result on sufficient conditions that guarantee that the entire BLCM is identifiable. Interestingly, all our conditions are neatly stated in terms of the bipartite graph $\Gamma$. The key concept is the following "double triangular" structure in $\Gamma$.

**Definition 3** (Triangular $\Gamma$-matrix). *A $K \times K$ matrix $\Gamma_1$ with binary entries is "triangular" when it can be written as the following form after arbitrary row and column permutations:*

$$\Gamma_1 = \begin{pmatrix} 1 & 0 & 0 & \ldots & 0 \\ * & 1 & 0 & \ldots & 0 \\ * & * & 1 & \ldots & 0 \\ \vdots & \vdots & \vdots & \ddots & \vdots \\ * & * & * & \ldots & 1 \end{pmatrix}. \tag{1}$$

*Here, each $* \in \{0, 1\}$ denotes an arbitrary value.*

*Additionally, we say that the $J \times K$ bipartite adjacency matrix $\Gamma$ is "double triangular" when it can be written (after possibly permuting the rows) as:*

$$\Gamma = \begin{pmatrix} \Gamma_1 \\ \Gamma_2 \\ \Gamma_3 \end{pmatrix}, \tag{2}$$

*where the $K \times K$ matrices $\Gamma_1, \Gamma_2$ are triangular (with potentially different column permutations). The remaining $(J - 2K) \times K$ matrix $\Gamma_3$ can be an arbitrary matrix, and may have zero rows.*

**Example 1.** *Still consider the graphical structure with $K = 3$ latent variables and $J = 8$ observed variables from Figure 1. Here, we verify that this structure is double triangular by explicitly defining the row permutation that gives (2). For example, if we define $\Gamma_1, \Gamma_2, \Gamma_3$ to correspond to the observed variables $(X_1, X_2, X_3), (X_6, X_7, X_5), (X_4, X_8)$, we get:*

$$\Gamma_1 = \begin{pmatrix} 1 & 0 & 0 \\ 1 & 1 & 0 \\ 1 & 0 & 1 \end{pmatrix}, \quad \Gamma_2 = \begin{pmatrix} 0 & 1 & 0 \\ 0 & 1 & 1 \\ 1 & 0 & 1 \end{pmatrix} \overset{permute\ columns}{\sim} \begin{pmatrix} 1 & 0 & 0 \\ 1 & 1 & 0 \\ 0 & 1 & 1 \end{pmatrix}, \quad \Gamma_3 = \begin{pmatrix} 1 & 1 & 1 \\ 0 & 0 & 0 \end{pmatrix}.$$

*It is immediate that $\Gamma_1$ is a $K \times K$ triangular matrix, and $\Gamma_2$ can also be written as a triangular matrix after permuting the columns.*

**Remark 2** (Comparison with the pure children condition). *The double triangular condition* (2) *generalizes the popular "two pure children per latent" condition that guarantees identifiability of various latent variable models (Xie et al., 2022; Chen et al., 2022; Lee & Gu, 2026). Note that an observed variable $X_j$ is called a* pure child *when it has exactly one latent parent $H_k$, or equivalently, the corresponding (jth) row in $\Gamma$ is a standard basis vector $e_k$. By setting all the arbitrary values indexed by $*$ in* (1) *equal to zero, we get $\Gamma_1 = \Gamma_2 = I_K$. This is equivalent to assuming two pure children per latent variable. Thus, the double triangular condition relaxes the two pure children condition by allowing a significantly denser bipartite graph $\Gamma$.*

*To illustrate this we consider applications in educational testing, where there are often complex skills that are measured alongside another skill, as opposed to having observed pure children. Various real-world test designs and estimated graphical structures satisfy the double-triangular condition, but violates the usual two pure children condition. For example, the test design for TOEFL reading (see Table A2 in von Davier (2008)) does not contain any pure children for the complex skill "Synthesize and organize", but this design still satisfies our double triangular condition. The same phenomenon can be observed for the estimated bipartite graph $\widehat{\Gamma}$ of a benchmark educational dataset as well (see Table 10 in Appendix F).*

**Remark 3** (Verifying the double triangular condition). *To verify the double triangular condition, one can take a greedy approach to sequentially search triangular sub-matrices. To elaborate, first find a standard basis row-vector $e_k$. For example, this is the bold row in the left term in the below equation. This becomes the first row in the triangular structure in* (1) *after row/column permutations. Let $\Gamma^{(1)}$ be the $(J-1) \times (K-1)$ matrix obtained from $\Gamma$ by removing the corresponding row and kth column (marked blue below).*

*Next, we search for a standard basis row-vector in $\Gamma^{(1)}$, which is computationally as easy as the step above. For example, this is the bold row in the right term of the below equation. This procedure searches the second row in* (1)*. This is because after removing the first column in* (1)*, the second row in* (1) *(of length $K-1$) must also be a standard basis vector. By continuing this procedure inductively, we can identify a triangular structure. We can repeat the same procedure for the remaining rows to search for the second triangular structure.*

$$\Gamma = \begin{pmatrix} 1 & 0 & 1 \\ \boldsymbol{0} & \boldsymbol{1} & \boldsymbol{0} \\ 0 & 1 & 1 \end{pmatrix} \xrightarrow{\text{remove 2nd row / col}} \Gamma^{(1)} = \begin{pmatrix} 1 & 1 \\ \boldsymbol{0} & \boldsymbol{1} \end{pmatrix}.$$

*In practice, the double-triangular condition can be verified in multiple ways. For example in Figure 1, the row-indices for the triangular matrices $\Gamma_1, \Gamma_2$ can be chosen as (a) $\{1,2,3\}, \{5,6,7\}$, (b) $\{1,3,7\}, \{2,5,6\}$, or (c) $\{1,5,7\}, \{2,3,6\}$. Thus, while the success of the above procedure may require multiple restarts in worst-case settings, it can practically verify the double-triangular condition.*

Now, we present our main identifiability results. We start with identifying the latent dimension $K$.

**Theorem 1** (Identifying $K$). *For each $j$, let $\mathcal{X}_j$ be a nondegenerate separable metric space. Suppose that the true $\Gamma$-matrix is double triangular. Then, the number of latent variables $K$ is identifiable.*

The proof argument is based on the following key property, which shows that a triangular matrix $\Gamma_1$ forces the corresponding conditional probability table to have full rank. We prove Lemma 1 in Appendix D, and Theorem 1 in Appendix C.

**Lemma 1.** *Consider a BLCM with binary responses, that is $\mathcal{X}_j = \{0,1\}$ for all $j$. Suppose that the $\Gamma$-matrix corresponding to $X_1, \ldots, X_K$ is lower-triangular and can be written as in* (1)*. Then, the $2^K \times 2^K$ conditional probability table $\mathbb{P}(X_{1:K} \mid H)$ has full column-rank.*

Here, for simplicity, we prove Theorem 1 assuming binary responses. The full proof under general responses builds upon discretizing the sample space $\mathcal{X}_j$, and is postponed to Appendix C.

*Simplified proof of Theorem 1.* As the true model is double triangular, let $S_1$ and $S_2$ denote two disjoint subsets of $[J]$ that index the rows of $\Gamma_1$ and $\Gamma_2$, respectively. Under the modeling assumptions and recalling

that $\pi := (\pi_h)_{h \in \{0,1\}^K}$ for $\pi_h = \mathbb{P}(H = h)$, we can write

$$\mathbb{P}(X_{S_1}, X_{S_2}) = \underbrace{\mathbb{P}(X_{S_1} \mid H)}_{2^{|S_1| \times 2^K}} \times \underbrace{\mathsf{diag}(\pi)}_{2^K \times 2^K} \times \underbrace{\mathbb{P}(X_{S_2} \mid H)^\top}_{2^K \times 2^{|S_2|}} \tag{3}$$

$$= \sum_{h \in \{0,1\}^K} \pi_h \mathbb{P}(X_{S_1} \mid H = h) \circ \mathbb{P}(X_{S_2} \mid H = h).$$

Here, $\circ$ denotes the outer product of vectors. Now, Lemma 1 implies that $\mathbb{P}(X_{S_1} \mid H), \mathbb{P}(X_{S_2} \mid H)$ have full column rank of $2^K$. Additionally, under part (a) of the nondegeneracy Assumption 3, the diagonal matrix $\mathsf{diag}(\pi)$ also has full rank $2^K$. Thus, $\mathbb{P}(X_{S_1}, X_{S_2})$ has rank $2^K$.

Now, consider an alternative model with $L < K$ latent variables (indexed by $\widetilde{G}, \widetilde{\mathbb{P}}$) that defines an identical marginal distribution for the observed variables $\widetilde{\mathbb{P}}(X) = \mathbb{P}(X)$. Then, writing $\widetilde{\pi}_h := \widetilde{\mathbb{P}}(\widetilde{H} = h)$, we must have

$$\mathbb{P}(X_{S_1}, X_{S_2}) = \widetilde{\mathbb{P}}(X_{S_1}, X_{S_2})$$

$$= \underbrace{\widetilde{\mathbb{P}}(X_{S_1} \mid \widetilde{H})}_{2^{|S_1| \times 2^L}} \times \underbrace{\mathsf{diag}(\widetilde{\pi})}_{2^L \times 2^L} \times \underbrace{\widetilde{\mathbb{P}}(X_{S_2} \mid \widetilde{H})^\top}_{2^L \times 2^{|S_2|}},$$

which gives $\mathsf{rk}(\mathbb{P}(X_{S_1}, X_{S_2})) \leq 2^L < 2^K$. Thus, we have a contradiction, and such an alternative model does not exist. $\qquad\square$

**Remark 4** (Allowing alternative models with more latent variables). *Note that Theorem 1 only identifies the lower bound of $K$, as we consider alternative models with "$\widetilde{K} < K$" in property (a) of Definition 2. This restriction for $\widetilde{K}$ can be relaxed (i.e., the precise value of $K$ can be identified), at the cost of considering alternative BLCMs that are double triangular. The proof argument is similar to Theorem 1, see Proposition 2 in Appendix C for a formal statement.*

Now, we move on to identifying the remaining model components. We show that the bipartite graph $\Gamma$ of a BLCM with given $K$ is identifiable when $\Gamma$ is double triangular. To identify the latent DAG $\Lambda$ as well as all conditional distributions $\mathbb{P}_{j,h}$ and latent proportions $\pi$, we additionally require the following subset condition. This condition was first proposed in Pearl & Verma (1992) for latent causal graphical models, and was also assumed in several recent works as well (Evans, 2016; Kivva et al., 2021).

**Definition 4** (Subset condition). *We say that the $J \times K$ matrix $\Gamma$ satisfies the subset condition if for different latent variables $H_k \neq H_l$, $\mathsf{ch}_X(H_k)$ is not a subset of $\mathsf{ch}_X(H_l)$ and vice versa; or equivalently, there exists no partial order between any two columns of $\Gamma$ indexed by $k \neq l$.*

The subset condition is not directly implied by the double triangular assumption, as the columns of $\Gamma$ may exhibit a partial order when all * values in (1) are set to 1, and both $\Gamma_1, \Gamma_2$ in (2) exhibit an identical triangular structure. However, the subset condition is satisfied when there exist a pure child for each latent variable, so this condition is weaker than assuming pure children.

**Theorem 2** (Identifiability of model components). *Consider a BLCM with a known latent dimension $K$. Assuming that (i) the true $\Gamma$-matrix is double triangular, and that (ii) all columns in $\Gamma_3$ of Definition 3 are not empty, $\Gamma$ is identifiable. Additionally, when the true $\Gamma$-matrix satisfies the subset condition (in addition to (i) and (ii)), the latent graph $\Lambda$ and probability distribution $\mathbb{P}$ are identifiable.*

*Proof sketch of Theorem 2.* For simplicity, suppose all sample spaces $\mathcal{X}_j$ are binary. Similar to earlier, let $S_1, S_2, S_3$ be the partition of $[J]$ such that each $S_a$ index the rows of $\Gamma_a$. Our key idea is to show that the following tensor CP decomposition (a three-way generalization of the matrix decomposition (3)) of the full marginal distribution $\mathbb{P}(X)$ is unique:

$$\mathbb{P}(X_{S_1}, X_{S_2}, X_{S_3}) = \left[\!\left[ \underbrace{\mathbb{P}(X_{S_1} \mid H) \times \mathsf{diag}(\pi)}_{2^{|S_1| \times 2^K}}, \; \underbrace{\mathbb{P}(X_{S_2} \mid H)}_{2^{|S_2| \times 2^K}}, \; \underbrace{\mathbb{P}(X_{S_3} \mid H)}_{2^{|S_3| \times 2^K}} \right]\!\right] \tag{4}$$

$$= \sum_{h \in \{0,1\}^K} \pi_h \mathbb{P}(X_{S_1} \mid H = h) \circ \mathbb{P}(X_{S_2} \mid H = h)$$
$$\circ \, \mathbb{P}(X_{S_3} \mid H = h).$$

See Lemma 3 for additional details and notations for tensor decompositions.

We separate the proof into three steps as follows.

*Step 1: Apply Kruskal's theorem to identify the components in* (4)*.* By Lemma 1, the first two components of (4) have full column rank. Also, Assumption 3 implies that all columns of the third component are distinct. This allows us to apply Kruskal's theorem for the uniqueness of tensor decompositions (Kruskal, 1977, also see Lemma 3). Hence, we can identify the three components of (4) up to a permutation of $2^K$ components.

*Step 2: Identify the bipartite graph* $\Gamma$*.* We identify the bipartite graph $\Gamma$ up to a permutation of its $K$ columns. The main idea is to partition the $2^K$ tensor component indices based on the corresponding columns of $\mathbb{P}(X_{S_2} \mid H)$ in (4), and resolve the label permutation based on the triangular structure of $\Gamma_2$. Then, we identify each row of $\Gamma$ via an induction on $k = 1, \ldots, K$.

*Step 3: Identify* $\pi, \mathbb{P}_{j,h}$*, and the latent DAG* $\Lambda$*.* Using the assumption that $\Gamma$ satisfies the subset condition, we map each tensor component in (4) to a binary vector representation, up to label permutation and sign flipping. Given this, $\pi$ and $\mathbb{P}_{j,h}$ are identified by marginalizing out each component of (4). Finally, using faithfulness (Assumption 2), the DAG $\Lambda$ can be identified up to its Markov equivalence class from $\pi$. □

**Remark 5** (Extension to general responses)**.** *The above proof argument generalizes to arbitrary observed variables that take values in any nondegenerate metric spaces (as stated at the beginning of Section 2) by discretizing the sample space. By choosing a wise discretization, the discretized observations still satisfy Assumption 3(b), and a similar argument using Kruskal's theorem can be applied to identify the DAGs and the latent proportion vector* $\pi$*. Identifying the conditional distribution itself is trickier, for which we use ideas from measure theory (separating classes) to identify the full conditional distributions (e.g., continuous densities) from the discretized p.m.f.*

Next, we revisit our running example in Figure 1 and illustrate that the conditions in Theorem 2 are easy to check.

**Example 2.** *Recall our running example from Figure 1 and Example 1. We have already verified that* $\Gamma$ *is double triangular, and the remaining matrix* $\Gamma_3 = \begin{pmatrix} 1 & 1 & 1 \\ 0 & 0 & 0 \end{pmatrix}$ *has no empty columns. Also,* $\Gamma$ *satisfies the subset condition since there is no partial order between any two columns. Thus, the BLCM parameters with a graphical structure as in Figure 1 is identifiable by Theorem 2.*

**Remark 6** (Discussion on the identifiability notion)**.** *While Theorem 2 imposes assumptions on the true model components, it establishes identifiability against* arbitrary alternative models $(\widetilde{G}, \widetilde{\mathbb{P}})$ *that only satisfy the minimal BLCM requirements in Definition 1. In other words, we do not establish identifiability by assuming that the alternative model also satisfies the double triangular condition. Note that this is more general than the notion of* recoverability *often used in causal discovery.*

**Remark 7** (Relaxing the subset condition)**.** *One may assume alternative conditions on behalf of the subset condition in Theorem 2. One possibility is to assume* monotonicity *of the latent variable* $H$*, by supposing*

$$\mathbb{P}_{j,h}(C_j) > \mathbb{P}_{j,h'}(C_j), \quad \forall h \neq h' \;\; s.t \;\; h_{\mathsf{pa}(X_j)} \succ h'_{\mathsf{pa}(X_j)}, \tag{5}$$

*for some fixed baseline set* $C_j \subsetneq \mathcal{X}_j$*. For example, one may take* $C_j = \{1\}$ *for binary responses, and* $C_j = (0, \infty)$ *for continuous responses. See the following proposition for a formal statement. While assuming monotonicity does restrict the parameter space, it additionally resolves the sign-flipping ambiguity as well as enhancing interpretability.*

**Proposition 1** (Modification of Theorem 2 under monotonicity)**.** *Consider a BLCM with a known latent dimension* $K$ *that satisfies the above monotonicity condition* (5)*. Then, the BLCM is identifiable under conditions (i), (ii) in Theorem 2.*

### 3.2 Necessary Condition

Next, we establish necessary conditions for identifiability. To summarize, we show that each column of $\Gamma$ must not be too sparse (Theorem 3) as well as not too dense (Theorem 4).

When all responses are categorical, trivial requirements for identifiability follow from comparing the number of equations and parameters. The following theorem strengthens this by showing that each latent variable must be measured by at least three observed variables. This is a fundamental requirement under categorical latent variables (beyond binary), and in fact we prove Theorem 3 under the general assumption of categorical latent variables $H$ (with known cardinality). This strengthens the usual requirement for two measurements per latent (e.g. see Lemma 3 in Evans, 2016).

**Theorem 3** (Three measurements per latent). *For a BLCM with known $K$ to be identifiable, it is necessary for each latent variable $H_k$ to have at least* three *observed children. This necessary condition also holds when the latent variables $H$ are categorical (i.e., polytomous and not binary).*

Theorem 3 justifies the decomposition of $\Gamma$ into three components as in (2). The sufficient condition in Theorem 2 requires all columns of $\Gamma_a$ to be non-empty, so each $\Gamma_a$ measures each latent variable $H_k$ at least once.

Our next necessity result proves the necessity of the subset condition (see Definition 4). This condition is closely related to the sign-flipping of $H_k = 0/1$ for each latent variable, and the following result shows its necessity even under a known graphical structure $\Gamma$.

**Theorem 4** (Subset condition). *For a BLCM with known $K \geq 2$ and known $\Gamma$ to be identifiable, the $\Gamma$-matrix must satisfy the subset condition.*

In the following two examples, we motivate each necessary condition in Theorems 3 and 4.

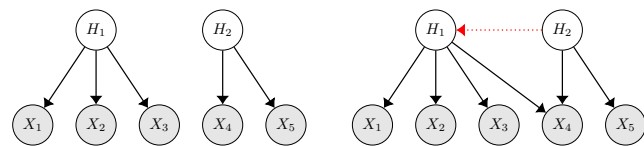

| $h = (h_1, h_2)$ | $(0,0)$ | $(0,1)$ | $(1,0)$ | $(1,1)$ |
|---|---|---|---|---|
| $\pi_h$ | 0.36 | 0.24 | 0.24 | 0.16 |
| $\widetilde{\pi}_h$ | 0.36 | 0.24 | 0.16 | 0.24 |

Figure 2: Counterexamples of three measurements per latent.

Table 2: Counterexample of the subset condition. The first/second row denotes true/alternative proportion parameters $\pi/\widetilde{\pi}$.

**Example 3** (Violation of three measurements per latent variable). *Suppose that the true model follows the DAG in the left panel of Figure 2, where the latent variable $H_2$ is measured only twice. For simplicity, suppose that the graphical structures $(\Gamma, \Lambda)$ are known and that all observed variables are binary. As $H_1$ and $H_2$ are independent, $(X_1, X_2, X_3) \perp H_2$. Thus, one has to identify the conditional distributions associated with $H_2$ only based on $\mathbb{P}(X_4, X_5)$. Simply comparing the number of effective parameters for this structure (5 parameters, as one parameter is required for $P(H_2)$, and four parameters are required for $P(X_j \mid H_2 = h), j = 4, 5, h = 1, 2$) versus the number of equations provided by the marginal distribution of $(X_4, X_5)$ (3 equations, by counting the number of configurations of $(X_4, X_5)$ and noting the p.m.f. constraint) demonstrates non-identifiability. On the contrary, the latent variable $H_1$ does not suffer from this issue, as it has three measurements as opposed to two. In fact, the conditional distributions $\mathbb{P}(X_j \mid H_1)$ for $j = 1, 2, 3$ are still identifiable.*

*Theorem 3 extends this non-identifiability argument to* arbitrary *latent structures such as that on the right panel of Figure 2, where dependent latent variables $H_1, H_2$ as well as additional latent parents of $X_4$ are allowed.*

**Example 4** (Violation of the subset condition). *Consider a BLCM with $K = 2$ latent variables that are independent: $H_1 \perp H_2$ (i.e. $\Lambda = \phi$) and $\mathbb{P}(H_1 = 1) = \mathbb{P}(H_2 = 1) = 0.4$. Suppose that we can write $\Gamma$ as in (2), where $\Gamma_1 = \Gamma_2 = \Gamma_3 = \begin{pmatrix} 1 & 0 \\ 1 & 1 \end{pmatrix}$. This $\Gamma$ satisfies all conditions in Theorem 2 except for the subset condition. Then, even though $\Gamma$ is identifiable by Theorem 2, the latent causal graph $\Lambda$ is non-identifiable. For this goal, the proof of Theorem 4 constructs an alternative distribution $\widetilde{\mathbb{P}}$ with $\widetilde{\pi}_h = \widetilde{\mathbb{P}}(H = h)$ as in the*

*second row of Table 2, and similarly define conditional probabilities $\widetilde{\mathbb{P}}_{j,h}$. Under $\widetilde{\mathbb{P}}$, $H_1$ and $H_2$ are dependent, so the latent DAG $\Lambda$ is non-identifiable.*

In Appendix B, we illustrate that the necessary conditions in Theorems 3 and 4 themselves do not guarantee identifiability, which justifies the gap between our necessary and sufficient conditions. Closing this gap is certainly important, but also known to be notoriously challenging. One exception is the finite mixture model with $L$ mixture components and $J$ binary observations, where $J \geq 2L - 1$ was recently shown to be the tight identifiability requirement (Tahmasebi et al., 2018; Lyu & Yang, 2025). However, under a weaker notion of "generic identifiability" that allows a measure-zero non-identifiable parameter space, Allman et al. (2009) showed a much weaker sufficient condition of $J \geq 2 \log_2 L + 1$ (whose tightness is unknown).

Interestingly, viewing our causal model as a finite mixture with $L = 2^K$ components, the double-triangular condition (more generally, Theorem 2) requires $J \geq 2K + 1 = 2 \log_2 L + 1$ observed variables. It is quite surprising that our non-generic result matches the generic result in Allman et al. (2009), and significantly relaxes the requirement $J \geq 2^{K+1} - 1$. Additionally, our lower bound on $J$ also relaxes that in other discrete causal discovery works, such as $J \geq 3K$ in Chen et al. (2024).

## 4 Experiments

We conduct experiments to empirically validate our identifiability results under finite samples. Building upon the identifiability guarantees of the entire model, we implement a score-based estimator by discretizing the continuous responses and maximizing the regularized log-likelihood. Our implementation first utilizes a penalized EM algorithm (Ma et al., 2023) to learn the bipartite graph $\Gamma$ alongside the latent proportion vector $\pi$. Then, we use the estimated latent proportions $\widehat{\pi}$ to recover the latent graph $\Lambda$. The implementation details are provided in Appendix E.2.

The primary purpose of the experiments is to better understand the consequences of identifiability results in simulated and real data. Using simulated data, we validate the identifiability result in Theorem 2 and also validate the necessary conditions in Theorems 3, 4. Finally, we provide a real-data illustration of the double-triangular condition. Throughout the first two simulated experiments, we assume that the number of latents $K$ is given, and focus on evaluating the causal graph recovery. We conduct additional experiments regarding selecting $K$ in Appendix E.3, where we validate Theorem 1.

**Experiments under varying latent structures**  Following Figure 1, assume a true BLCM with $K = 3$ latent variables and $J = 8$ observed variables, where $X_1, X_2, X_3, X_4$ are binary and $X_5, X_6, X_7, X_8$ are continuous. We assume that $X_5, X_6$ are generated from a Normal and that $X_7, X_8$ are generated from a Cauchy distribution. Note that the distributional assumptions are not used for estimation. In terms of the graphical structures, $\Gamma$ is defined as in Figure 1, and we consider three settings for $\Lambda$: (a) chain $(H_1 \to H_2 \to H_3)$, (b) collider $(H_1 \to H_2 \leftarrow H_3)$, (c) all dependent (as in Figure 1). We consider varying sample sizes of $N = 1000, 5000, 10000$, and conduct 300 independent simulation trials for each setting.

| $\Lambda \setminus N$ | $\mathrm{SHD}(\widehat{\Gamma}, \Gamma)$ | | | $\mathrm{SHD}(\widehat{\Lambda}, \Lambda)$ | | |
|---|---|---|---|---|---|---|
| | 1k | 5k | 10k | 1k | 5k | 10k |
| Chain | 1.90 | 1.64 | 1.48 | 0.57 | 0.46 | 0.38 |
| Collider | 2.51 | 2.24 | 2.05 | 0.36 | 0.27 | 0.24 |
| Dependent | 1.83 | 1.56 | 1.35 | 0.45 | 0.41 | 0.35 |

Table 3: SHD for estimating $\Gamma$ and $\Lambda$, **smaller is better**. Maximum error for $\Gamma$, $\Lambda$ is $24, 3$, respectively.

In Table 3, we report the average structural Hamming distance (SHD) between the estimated and true DAG, separately for $\Gamma$ and $\Lambda$. The SHD computes the number of incorrectly estimated edges between the two graphs by comparing the distance between the equivalence classes of PDAGs (Tsamardinos et al., 2006). The results in Table 3 illustrate that both $\Gamma$ and $\Lambda$ can be effectively estimated regardless of the true latent

structure $\Lambda$, and that the accuracy increases as the sample size $N$ increases. This clearly validates that latent structures can be accurately recovered under our weak identifiability conditions. See Figure 3 in Appendix E for boxplots corresponding to Table 3.

**Ablation studies** We conduct ablation studies and consider settings when the conditions in Theorem 2 does not hold. In particular, we consider two alternate bipartite graphical structures where each necessary condition in Theorems 3 and 4 are violated. Here, $\Gamma^{\mathrm{DT}}$ is the 'double-triangular' bipartite graphical structure in Figure 1 and satisfies our identifiability conditions in Theorem 2. $\Gamma^{\mathrm{dense}}$ is a denser bipartite graph, where the subset condition does not hold (i.e. contradicting the necessary condition in Theorem 3). $\Gamma^{\mathrm{sparse}}$ is a sparser bipartite graph, where the three-measurement-per-latent condition (see Theorem 3) does not hold. See (13) in Appendix E.1 for the explicit forms.

| $\Gamma \setminus N$ | $\mathrm{SHD}(\widehat{\Gamma}, \Gamma)$ | | | $\mathrm{SHD}(\widehat{\Lambda}, \Lambda)$ | | |
|---|---|---|---|---|---|---|
| | 1k | 5k | 10k | 1k | 5k | 10k |
| $\Gamma^{\mathrm{DT}}$ | 1.90 | 1.64 | 1.48 | 0.57 | 0.46 | 0.38 |
| $\Gamma^{\mathrm{dense}}$ | 4.62 | 3.58 | 3.01 | 1.46 | 1.59 | 1.66 |
| $\Gamma^{\mathrm{sparse}}$ | 3.95 | 4.04 | 4.00 | 0.81 | 0.76 | 0.77 |

Table 4: SHD under varying $\Gamma$ and $N$, where $\Lambda$ is fixed as the chain graph. Only $\Gamma^{\mathrm{DT}}$ satisfy the proposed identifiability conditions.

The results in Table 4 reveal performance degradation for the theoretically non-identifiable cases. The error in these non-double-triangular cases does not decrease as the sample size $N$ increases. This suggests the fundamental failure of identifiability even at the population level, and empirically verifies our necessary conditions.

Table 5 reports more detailed entrywise estimation accuracy of the $8 \times 3$ bipartite graph $\Gamma$ under the setting of Table 4 with $N = 10000$. Instead of reporting the aggregated SHD metric, we report the entrywise empirical average of $I(\widehat{\gamma}_{j,k} \neq \gamma_{j,k})$ under each $\Gamma$. In particular, for the dense case, the subset condition between the second and third columns (highlighted in bold) does not hold, which results in several large error values in these two columns. For the sparse case, the latent variable $H_3$ only has two observed children and violates Theorem 3, leading to a non-trivial error for the entire third column. These results illustrate that theoretical non-identifiability results in an accuracy drop for the graphical structure associated with non-identifiable latent variables.

| $\Gamma$ | $\Gamma^{\mathrm{DT}}$ | | | $\Gamma^{\mathrm{dense}}$ | | | $\Gamma^{\mathrm{sparse}}$ | | |
|---|---|---|---|---|---|---|---|---|---|
| $j \setminus k$ | 1 | 2 | 3 | 1 | **2** | **3** | 1 | 2 | **3** |
| 1 | | | | | | | | 0.16 | 0.23 |
| 2 | | | | 0.36 | | 0.70 | | | 0.27 |
| 3 | | 0.22 | | | 0.36 | | | 0.24 | 0.28 |
| 4 | | | 0.41 | | | 0.73 | | | 0.40 |
| 5 | | | 0.13 | | 0.17 | 0.23 | | 0.26 | 0.33 |
| 6 | | | | 0.24 | 0.51 | | 0.23 | | 0.30 |
| 7 | 0.18 | 0.17 | | 0.29 | 0.57 | | 0.18 | 0.56 | 0.12 |
| 8 | | | | | | | | 0.12 | 0.14 |

Table 5: Entry-wise estimation error for the bipartite graph $\Gamma$, where three $\Gamma$s are considered. The bold columns correspond to theoretically non-identifiable latent variables. Values $< 0.1$ are omitted, and larger values mean worse accuracy.

**Real data analysis** Finally, we analyze a benchmark educational testing dataset on fraction subtraction (Tatsuoka, 2002), which consists of binary responses that record students' correct/incorrect answers. Estimating the causal graph $\Gamma$ between the observed and latent variables has been an important problem in

educational measurement, and this dataset has been extremely popular due to the explicit nature of the questions (e.g. $\frac{3}{4} - \frac{3}{8}$ or $3\frac{1}{2} - 2\frac{3}{2}$).

By fitting the model with $K = 2, 3, 4$ latent variables, we chose $K = 3$ latent variables based on BIC. Based on the estimated bipartite graph $\widehat{\Gamma}$ (see Table 10 in the appendix), we can interpret the three latent skills as: computing common denominator, writing integer as fraction, and subtracting integers. Notably, the estimated $\widehat{\Gamma}$ does not have two pure children per latent, but satisfies the double triangular condition in Theorem 2. This illustrates the practical usefulness of our proposed conditions. Additionally, the estimated latent graph $\widehat{\Lambda}$ is fully dependent, which illustrates that all skills are closely correlated.

## 5    Discussion

This work opens up many directions for future work. First, it would be interesting to consider more complex graphical structures that go beyond measurement models under additional (but still weak) assumptions. For example, one may allow direct causal relationships between the observed variables or allow hierarchical latent variables, under additional structural constraints. Second, it would be important to generalize our identifiability conditions for causal models with general types of latent variables. This includes considering categorical latents (that may not be binary), as well as potentially allowing both categorical and continuous latents. In particular, for the case of categorical latents, we conjecture that our sufficient identifiability conditions may be extended at the cost of modifying Assumption 3(b) to a stronger "full-rank" requirement for the conditional probabilities $\mathbb{P}(X_j \mid \mathsf{pa}(X_j))$. Note that extending the score-based estimator in Section 4 to categorical latents is not a bottleneck, since extending the penalized likelihood estimation strategy is immediate. Finally, it would be important to develop a nonparametric method that can fully leverage continuous responses to estimate individual conditional distributions, for example using kernel methods.

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

## Appendix

This Appendix is organized as follows. Section A discusses three closely related works in-depth. Section B provides additional examples. Section C proves all main theorems, and Section D proves all technical lemmas. Section E provides all implementation details for our simulations alongside comparison studies. Section F provides details for real data analysis, and Section G provides detailed theoretical and practical justifications regarding our setup with binary latent variables. The code implementation is available as a online supplement in https://openreview.net/forum?id=KiiSlAsLuN.

## A  Detailed literature review

**Comparison to Kivva et al. (2021)**  Both our work and Kivva et al. (2021) consider discrete latent variables without any assumptions on the true latent structure. While we require binary latent variables, Kivva et al. (2021) allow arbitrary categorical latent variables with unknown number of categories. However, their arguments rely on the existence of a mixture oracle, which gives the number of mixture components of each marginal distribution $\mathbb{P}(X_S)$ for $S \subset [J]$. This does not hold under our minimal model definition. For example, for binary responses with $\mathcal{X}_1 = \{0, 1\}$, the one-dimensional marginal $\mathbb{P}(X_1 = 1)$ does not contain enough information and its mixture components are fundamentally non-identifiable. The same argument holds for continuous responses as well, unless we impose parametric assumptions or separation between the conditional distributions $\mathbb{P}_{1,h}$.

One may point out that to identify the bipartite graph $\Gamma$, Kivva et al. (2021) only utilizes the mixture oracle to obtain the number of mixture components for three observed variables, say $X_S = (X_1, X_2, X_3)$. Without further assumptions, the mixture oracle does not exist for this goal as well. To see this, consider a simple setting with $K = 2$ and $J = 3$, where $H_1 \to \{H_2, X_1, X_2\}, H_2 \to X_3$. Then, the minimal number of mixture components in $(X_1, X_2, X_3)$ is 2 because $X_1, X_2, X_3$ are conditionally independent given $H_1$. Hence, under our setup, it is impossible to conclude that there must be $2 \times 2 = 4$ mixture components, which corresponds to all configurations of $(H_1, H_2)$.

Another point of comparison is regarding the use of tensor decompositions. Kivva et al. (2021) also utilizes tensor decompositions to identify the bipartite graph $\Gamma$, but looks at the three-way tensor whose $(j_1, j_2, j_3)$-th element corresponds to the (log) *number of mixture components* of the distribution $(X_{j_1}, X_{j_2}, X_{j_3})$. Computing the value of this tensor requires a mixture oracle. On the other hand, our three-way tensor arises from re-shaping the marginal probability mass function (pmf) $\mathbb{P}(X_1, \ldots, X_J) = \mathbb{P}\Big((X_1, \ldots, X_K), (X_{K+1}, \ldots, X_{2K}), (X_{2K+1}, \ldots, X_J)\Big)$ based on the double triangular structure. Hence, our tensor does not require any additional information beyond that in the observed data $X$.

A final point of comparison is about learning $\Lambda$. While the proof strategy (especially reducing the ambiguity of latent variable value up to sign flip) shares the similar spirit, the key distinction arises again from the mixture oracle. Kivva et al. (2021) utilizes the number of mixture components to construct a projection mapping between the mixture components of $X_S$ and the mixture components of each mode $X_j$. As this information is not available in our work, we take a detour and utilize the sparsity patterns in the conditional distributions $\mathbb{P}_{j,h}$ instead.

**Comparison to Chen et al. (2024; 2025)**  Both our work and Chen et al. (2024; 2025) establish identifiability by utilizing tensor decompositions of the observed variables. In particular, Chen et al. (2025) allows both discrete and continuous responses by using the key idea of discretizing the continuous responses, similar to our paper. While both works can allow general categorical latent variables, they require stronger identifiability conditions compared to this paper.

First, both works require three pure children per latent variable, as opposed to our double triangular condition. Second, they require the cardinality of the observed variables $|\mathcal{X}_j|$ to be *strictly* greater than the cardinality of any latent variable. This does not allow binary responses as opposed to our result. Third, they operate under a stronger nondegeneracy assumption that requires full-rankness of the conditional probability table

$\mathbb{P}(X_j \mid \mathsf{pa}(X_j))$. In contrast, we do not require any full rank assumptions and assume a minimal nondegeneracy condition (see Assumption 3).

We also mention that one of their main contribution is on algorithmic recoverability based on a novel tensor rank criteria, and they do not establish identifiability by considering alternative models that do not contain pure children (in the sense of Remark 6).

**Comparison to Lee & Gu (2026)**  The works Lee & Gu (2024b; 2026) also establishes identifiability of binary latent variable models for arbitrary responses. As our proof argument for establishing nonparametric identifiability is motivated by this line of work, it is important to clarify the differences. Since Lee & Gu (2026) extends the conclusion of Lee & Gu (2024b), which requires the bipartite graph $\Gamma$ to be known, we mainly focus on comparing our work with Lee & Gu (2026).

First, the strict identifiability result in Lee & Gu (2026) require two pure children per latent variable, and mainly assumes the conditional distributions $\mathbb{P}_{j,h}$ to be a parametric generalized linear model:

$$\mathbb{P}_{j,h} \overset{d}{\equiv} \mathsf{ParFam}(\beta_{j,0} + \sum_{k=1}^{K} \beta_{j,k} h_k, \gamma_j), \quad \forall j, h.$$

Under the weaker notion of "generic" identifiability that allows a measure-zero subset of the true parameter space to be non-identifiable, they establish weaker identifiability conditions that do not require any pure children. We note that this condition is even weaker than our double triangular criteria, as they do not require any triangular structure. By assuming a multi-layer architecture, they additionally establish identifiability of hierarchical latent structures.

In comparison to the strict identifiability result in Lee & Gu (2026), we do not require any pure children nor any parametric structure. The generic identifiability result there also requires the generalized linear model parametric form, and is not able to pinpoint the non-identifiable true parameter values. In contrast, our results do not allow any sort of non-identifiability. Additionally, the theoretical results in Lee & Gu (2026) treats the latent dimension $K$ as known, whereas we establish its identifiability. Finally, we consider different latent architectures, as Lee & Gu (2026) considers a multi-layer latent structure without any between layer edges, but we consider a one-layer latent structure with arbitrary dependencies between the latent variables.

## B   Examples

Our first example illustrates the necessity of condition (b) in Assumption 3.

**Example 5** (Degenerate conditional distributions)**.** *For simplicity, suppose that the responses are binary. We construct an example with degenerate conditional distributions that lead to a non-identifiable $\Gamma$ matrix. For each $k \in [K]$, assume that the corresponding $\Gamma$ is lower triangular with all ones:*

$$\Gamma = \begin{pmatrix} 1 & 0 & 0 & \dots & 0 \\ 1 & 1 & 0 & \dots & 0 \\ 1 & 1 & 1 & \dots & 0 \\ \vdots & \vdots & \vdots & \ddots & \vdots \\ 1 & 1 & 1 & \dots & 1 \end{pmatrix}.$$

*For constants $a \neq b \in [0,1]$, suppose that the conditional distributions $\mathbb{P}_{k,h} := Ber(\theta_{k,h})$ are defined as:*

$$\theta_{k,h} = \begin{cases} a & \text{if } d_H(h, 0) \text{ is even,} \\ b & \text{if } d_H(h, 0) \text{ is odd.} \end{cases}$$

*Here, $d_H$ denotes the Hamming distance between two binary vectors. The conditional distributions are clearly degenerate, as $\theta_{k,h}$ can take only two possible values regardless of the configuration $h$, for all $k$.*

*Then, we can construct an alternative labeling $\widetilde{H}$ that corresponds to $\widetilde{\Gamma} = I_K$. See the following table for an illustration with $K = 3$. This illustrates the fundamental ambiguity resulting from nondegeneracy.*

| $j \setminus h$ | (0,0,0) | (1,0,0) | (0,1,0) | (1,1,0) | (0,0,1) | (1,0,1) | (0,1,1) | (1,1,1) |
|---|---|---|---|---|---|---|---|---|
| 1 | a | b | a | b | a | b | a | b |
| 2 | a | b | b | a | a | b | b | a |
| 3 | a | b | b | a | b | a | a | b |
| $\widetilde{h}$ | (0,0,0) | (1,1,1) | (0,1,1) | (1,0,0) | (0,0,1) | (1,1,0) | (0,1,0) | (1,1,1) |

Table 6: A degenerate pmf table of $\theta_{j,h} = \mathbb{P}(X_j = 1 \mid H = h)$, which can define multiple graphical structures: $\Gamma = \begin{pmatrix} 1 & 0 & 0 \\ 1 & 1 & 0 \\ 1 & 1 & 1 \end{pmatrix}$ and $\begin{pmatrix} 1 & 0 & 0 \\ 0 & 1 & 0 \\ 0 & 0 & 1 \end{pmatrix}$.

Note that under further degeneracy specifications such as the conjunctive assumption in Lee & Gu (2024a), it is possible to identify degenerate causal graphical models.

Our final example illustrates the gap between the necessary and sufficient conditions.

**Example 6** (Gap between necessary and sufficient conditions). *Consider a toy model with $K = 3$ latent variables and $J = 4$ binary observed variables, where the true bipartite graph is*

$$\Gamma = \begin{pmatrix} 1 & 1 & 0 \\ 1 & 0 & 1 \\ 0 & 1 & 1 \\ 1 & 1 & 1 \end{pmatrix}.$$

*This $\Gamma$ satisfies both the three measurement per latent condition (see Theorem 3) and the subset condition (see Theorem 4). However, simply observing that this model has 39 parameters (7 parameter for $\pi$ and 32 for the conditional distributions $\mathbb{P}_{j,h}$) but only 15 equations are given by the marginal distribution $\mathbb{P}(X)$, we can deduce that the model is non-identifiable. This illustrates that, without the double triangular condition to guarantee a sufficient number of observed variables, the model may be non-identifiable.*

**Remark 8** (The necessary conditions can be weakened under additional assumptions). *Our necessary conditions are stated under our minimal, nonparametric modeling framework. Under additional assumptions such as linearity and pure-children structures, the model can still be identifiable when there are two measurements per latent variable (Xie et al., 2020; Gu, 2025). Also, the subset condition is not required under (generalized) linear parametrizations (Theorem 2, Lee & Gu, 2026).*

## C Proof of main results

### C.1 Rigorous definition of trivial ambiguities

Before proving individual results, we formalize the three notions of ambiguities from Section 2.2. Recall from Section 2.3 that $\pi_h = \mathbb{P}(H = h)$ and $\mathbb{P}_{j,h}$ denotes the conditional distribution of $X_j$ given $H = h$.

**Definition 5.** *Consider two BLCMs with a known $K$ and model components $\big(\Lambda, \Gamma, \pi, \{\mathbb{P}_{j,h}\}\big)$ and $\big(\widetilde{\Lambda}, \widetilde{\Gamma}, \widetilde{\pi}, \{\widetilde{\mathbb{P}}_{j,h}\}\big)$, respectively. We formally say that these two models are equivalent up to* label permutation *when there exists a permutation $\sigma : [K] \to [K]$ such that for $\widetilde{h}_{h,\sigma} := (h_{\sigma_{(1)}}, \ldots, h_{\sigma(K)})$, we have*

$$\pi_h = \widetilde{\pi}_{\widetilde{h}_{h,\sigma}}, \quad \gamma_{j,k} = \widetilde{\gamma}_{j,\sigma(k)}, \quad \mathbb{P}_{j,h} = \widetilde{\mathbb{P}}_{j,\widetilde{h}_{h,\sigma}}, \quad \forall h \in \{0,1\}^K, \ j \in [J], \ k \in [K],$$

$$\{k \to \ell\} \in \Gamma \iff \{\sigma(k) \to \sigma(\ell)\} \in \widetilde{\Gamma}, \quad \forall k \neq \ell \in [K].$$

*Next, we say that the two models are equivalent up to* sign-flip *when* $\Lambda = \widetilde{\Lambda}, \Gamma = \widetilde{\Gamma}$ *and there exists permutations* $\tau_k : \{0,1\} \to \{0,1\}$ *for each* $k \in [K]$ *that satisfy*

$$\pi_h = \widetilde{\pi}_{(\tau_1(h_1),...,\tau_K(h_K))}, \quad \mathbb{P}_{j,h} = \widetilde{\mathbb{P}}_{j,(\tau_1(h_1),...,\tau_K(h_K))}, \quad \forall h \in \{0,1\}^K, j \in [J].$$

*Finally, we say that the two models are equivalent up to* Markov equivalence *when* $\pi = \widetilde{\pi}, \Gamma = \widetilde{\Gamma}, \mathbb{P} = \widetilde{\mathbb{P}}$, *and the latent graphs* $\Lambda$ *and* $\widetilde{\Lambda}$ *induce the same conditional independence relations (Pearl, 2014). Note that the bipartite graph* $\Gamma$ *does not suffer from Markov equivalence due to Assumption 1, which restricts the direction of the edges in* $\Gamma$.

## C.2 Proof of Theorem 1 and Proposition 2

*Proof of Theorem 1 for general responses.* We proceed in a similar spirit as in the sketch in the main text, where technical challenges arise from the arbitrary response types. Here, we reduce the responses to binary responses noting that the rank information is still preserved. For each $j$, let $C_j \subsetneq \mathcal{X}_j$ be a non-empty subset of the sample space. Let $Y_j$ be a binary random variable defined by setting

$$Y_j := I(X_j \in C_j). \tag{6}$$

Note that this implies

$$\mathbb{P}(Y_j = 1 \mid H = h) = \mathbb{P}(X_j \in C_j \mid H = h).$$

Then, by assumption Assumption 3, we have $\mathbb{P}(Y_j = 1 \mid H = h) \neq \mathbb{P}(Y_j = 1 \mid H = h')$ for any $h \neq h'$ such that $h_{\mathsf{pa}(X_j)} \neq h'_{\mathsf{pa}(X_j)}$. Hence, the second part of Assumption 3 holds for the discretized random variable $Y_j$.

Now, the argument in the main text can be applied by changing the notation $X$ to $Y$, and the proof is complete. $\square$

Next, we present the delegated statement (from Remark 4) for identifying $K$ under the class of BLCMs, where alternate models with more than $K$ latent variables are allowed.

**Proposition 2.** *Among the class of double triangular BLCMs, $K$ is identifiable in the weaker sense that the lower-dimensional restriction "$\widetilde{K} < K$" in condition (a) of Definition 2 can be relaxed to "$\widetilde{K} \neq K$".*

*Proof of Proposition 2.* Consider an alternative model with $L > K$ latent variables, that is double triangular and defines an identical marginal distribution $\widetilde{\mathbb{P}}(X) = \mathbb{P}(X)$. For each $j$, define the discretized random variable $Y_j$ as in (6). By the double triangular condition and (3), there exists some disjoint set of observed variables $X_{S_1}, X_{S_2} \subset (X_1, \ldots, X_J)$ such that the discretized pmf $\widetilde{\mathbb{P}}(Y_{S_1}, Y_{S_2})$ has full rank of $2^L > 2^K$. However, since the true model has $K$ latent variables, we can write

$$\mathbb{P}(Y_{S_1}, Y_{S_2}) = \mathbb{P}(Y_{S_1} \mid H)\mathsf{diag}(\pi)\mathbb{P}(Y_{S_2} \mid H)^\top,$$

which gives $\mathsf{rk}(\mathbb{P}(Y_{S_1}, Y_{S_2})) \leq 2^K$. Thus, we have a contradiction.

Now, the proof is complete, since Theorem 1 has already established that there exists no alternative model with $L < K$ latent variables. $\square$

## C.3 Proof of Theorem 2 and Proposition 1

Next, we show Theorem 2 and Proposition 1. Our first tool is the following technical lemma, which allows us to boil down the identifiability of models with arbitrary types of random variables to that with categorical variables. We need to use the notion of separable metric spaces and separating classes from standard probability textbooks (e.g. Chapter 1 of Billingsley (2013)).

**Lemma 2.** *For any separable metric space $\mathcal{X}_j$, there exists a countable separating class $\mathcal{C}_j$ whose values determine the probability measure on $\mathcal{X}_j$.*

The existence of such a separating class $\mathcal{C}_j$ is a consequence of $\mathcal{X}_j$ being a separating metric space, which was assumed in the first paragraph of Section 2. For a proof, see Step 1 in the proof of Theorem 1 in Lee & Gu (2024b). For example, for binary responses with $\mathcal{X}_j = \{0, 1\}$, we can simply take $\mathcal{C}_j = \{\{0\}, \{0, 1\}\}$. For continuous responses with $\mathcal{X}_j = \mathbb{R}$, we can take $\mathcal{C}_j = \{(-\infty, q) : q \in \mathbb{Q}\} \cup (-\infty, \infty)$, where $\mathbb{Q}$ is the set of rational numbers.

Having reduced the response types to categorical, our second tool is the celebrated Kruskal's theorem (Kruskal, 1977; Rhodes, 2010), which guarantees the unique CP decomposition of a three-way tensor. Before presenting the result, define the Kruskal rank of a $n \times r$ matrix $T_1$ (denoted as $\mathsf{rk}_k(T_1)$) as the largest number $R \leq r$ such that every distinct $R$ columns of $T_1$ are linearly independent. Also, for vectors $t_1, t_2, t_3$, denote their outer product as $t_1 \circ t_2 \circ t_3$.

**Lemma 3** (Kruskal's Theorem). *For $a = 1, 2, 3$, let $T_a$ be a $n_a \times r$ matrix, whose $\ell$th column is $t_{a,\ell}$. Let $\mathcal{T} := [\![T_1, T_2, T_3]\!] = \sum_{l=1}^{r} t_{1,\ell} \circ t_{2,\ell} \circ t_{3,\ell}$ be a three-way tensor with dimension $n_1 \times n_2 \times n_3$. Then, when the following condition holds:*

$$\mathsf{rk}_k(T_1) + \mathsf{rk}_k(T_2) + \mathsf{rk}_k(T_3) \geq 2r + 2,$$

*the rank $r$ decomposition of $\mathcal{T}$ is unique up to a common column permutation and rescaling of $T_a$s. In other words, if $\mathcal{T} := [\widetilde{T}_1, \widetilde{T}_2, \widetilde{T}_3]$ for some $n_a \times r$ matrices $\widetilde{T}_a$, there exists a permutation matrix $\Pi$ and invertible diagonal matrices $D_1, D_2, D_3$ with $D_1 D_2 D_3 = I_r$ such that $\widetilde{T}_a = T_a D_a \Pi$.*

*Proof of Theorem 2.* Consider a BLCM with a known $K$ and model parameters $(\Gamma, \Lambda, \mathbb{P})$ where $\Gamma$ satisfy the conditions in the Theorem statement (double triangular, $\Gamma_3$ does not have empty columns, subset condition). Suppose that there exists alternative model parameters $\widetilde{\Gamma}, \widetilde{\Lambda}, \widetilde{\mathbb{P}}$ that define the same marginal distribution $\mathbb{P}(X) = \widetilde{\mathbb{P}}(X)$. We show that $(\widetilde{\Gamma}, \widetilde{\Lambda}, \widetilde{\mathbb{P}})$ must be equal to $(\Gamma, \Lambda, \mathbb{P})$ up to index/value permutation and Markov equivalence. We separate the proof into three parts, after introducing the necessary notations.

*Step 0:* Additional notations regarding discretization.

We define the following notations under model parameters $(\Gamma, \Lambda, \mathbb{P})$. Assume analogous definitions under the alternative parameters $(\widetilde{\Gamma}, \widetilde{\Lambda}, \widetilde{\mathbb{P}})$.

For each $j \in [J]$ and $h \in \{0, 1\}^k$, let $\mathbb{P}_{j,h}$ be the conditional distribution of $X_j \mid H = h$:

$$\mathbb{P}_{j,h}(C) := \mathbb{P}(X_j \in C \mid H = h), \quad \forall C \subseteq \mathcal{X}_j. \tag{7}$$

For each $j \in [J]$, fix an integer $\kappa_j \geq 2$, and consider distinct measurable subsets $C_{1,j}, \ldots, C_{\kappa_j,j} \in \mathcal{C}_j$ with $C_{\kappa_j,j} = \mathcal{X}_j$. For each $j \in [J]$, define a $\kappa_j \times 2^K$ matrix $M_j$ by setting

$$M_j(l_j, h) := \mathbb{P}_{j,h}(C_{l_j,j}) = \mathbb{P}(X_j \in C_{l_j,j} \mid H = h).$$

Note that $C_{\kappa_j,j} = \mathcal{X}_j$ implies that $M_j(\kappa_j, h) = 1$ for all $h$, so the last row of $M_j$ is all-one.

Since the true model is double triangular, let $S_1$ and $S_2$ denote two disjoint subsets of $[J]$ that index the rows of $\Gamma_1$ and $\Gamma_2$. Also, let $S_3 := [J] \setminus (S_1 \cup S_2)$ denote the row indices of $\Gamma_3$. For each $a = 1, 2, 3$, set $\eta_a := \prod_{j \in S_a} \kappa_j$, and note that $\eta_1, \eta_2 \geq 2^K$. For each $a = 1, 2, 3$, let $T_a$ be a $\eta_a \times 2^K$ matrix defined as

$$T_a\big((l_j : j \in S_a), h\big) := \mathbb{P}(X_j \in C_{l_j,j}, \ \forall j \in S_a \mid H = h) = \prod_{j \in S_a} M_j(l_j, h).$$

Here, each row of $T_a$ is indexed by a vector $(l_j : j \in S_a)$, where $l_j \in [\kappa_j]$ for all $j$. Also, note that $T_a\big((\kappa_j : j \in S_a), h\big) = 1$ for all $h$.

Finally, define a $\eta_1 \times \eta_2 \times \eta_3$ tensor $\mathcal{T}$ by setting

$$\mathcal{T}\Big((l_j : j \in S_1), (l_j : j \in S_2), (l_j : j \in S_3)\Big) := \mathbb{P}(X_j \in C_{l_j,j}, \ \forall j \in [J])$$

$$= \sum_h \pi_h \prod_{a=1}^{3} T_a((l_j : j \in S_a), h).$$

Note that $\mathcal{T}$ merely reshapes the discretization of the marginal distribution $\mathbb{P}(X)$, and is identical under true and alternative model parameters.

Using these notations, we can generalize (4) in the main paper and write

$$\mathcal{T} = \left[\!\left[ T_1 \mathsf{diag}(\pi), T_2, T_3 \right]\!\right] = \left[\!\left[ \widetilde{T}_1 \mathsf{diag}(\widetilde{\pi}), \widetilde{T}_2, \widetilde{T}_3 \right]\!\right]. \tag{8}$$

*Step 1:* Apply Kruskal's theorem to identify the components in (8).

By Lemma 1, we have $\mathsf{rk}(T_1), \mathsf{rk}(T_2) = 2^K$. Note that even though $T_1, T_2$ are allowed to have more than $2^K$ rows, they still have full column rank, and consequently full Kruskal rank:

$$\mathsf{rk}_k(T_1), \mathsf{rk}_k(T_2) = 2^K.$$

For any binary vectors $h \neq h'$, there exists some $k \leq K$ such that $h_k \neq h'_k$. Recalling that $\Gamma_3$ has no empty columns, there exists some $j_k \in S_3$ such that $\gamma_{j_k,k} = 1$. In other words, $X_{j_k}$ measures $H_k$. Then, part (b) of Assumption 3 implies that $\mathbb{P}_{j_k,h} \neq \mathbb{P}_{j_k,h'}$, so the $h$-th and $h'$-th column of $T_3$ are distinct. Recalling that the last row of $T_3$ is 1 in every column, every two columns in $T_3$ are linearly independent, and we have

$$\mathsf{rk}_k(T_3) \geq 2.$$

Hence, the inequality in Lemma 3 holds, and Kruskal's theorem can be applied. This implies that the tensor decomposition (8) is unique in the following sense: there exists some $2^K \times 2^K$ permutation matrix $\Pi$ such that

$$\mathsf{diag}(\widetilde{\pi}) = \mathsf{diag}(\pi)\Pi, \quad \widetilde{T}_1 = T_1\Pi, \quad \widetilde{T}_2 = T_2\Pi, \quad \widetilde{T}_3 = T_3\Pi. \tag{9}$$

Note that the diagonal matrices $D_a$ in Lemma 3 can be omitted since the last rows of $T_a$ and $\widetilde{T}_a$ are all-one.

The conclusion (9) implies

$$\widetilde{M}_j = M_j\Pi, \quad \forall j \in [J],$$

where $\widetilde{M}_j, M_j$ are $\kappa_j \times 2^K$ matrices that describe the conditional distribution $\mathbb{P}_{j,h}$. For notational convenience, view the matrix $\Pi$ as a permutation mapping $\sigma_\Pi : \{0,1\}^K \rightarrow \{0,1\}^K$ that maps the column index $h$ to $\widetilde{h} = \sigma_\Pi(h)$. As $\sigma_\Pi$ can be an arbitrary permutation, we still need to structure it so that $h$ and $\sigma_\Pi(h)$ are equivalent up to label permutation and sign-flipping.

*Step 2:* Identify the bipartite graph $\Gamma$.

Using the triangular structure of the rows, we identify the $\widetilde{\Gamma}$ matrix up to label permutation. Our main idea is to again utilize the triangular structure of $\Gamma_2$ to first determine the label permutation, and then identify each row of $\widetilde{\Gamma}$. Recall that $S_2$ denotes the row indices that correspond to $\Gamma_2$ (assumed to be ordered as in (1)). Write $S_2 = \{j_1, \ldots, j_K\}$ so that the $j_k$-th row of $\Gamma$ is $\gamma_{j_k} = (\underbrace{*, \ldots, *}_{k-1}, 1, 0, \ldots, 0)$.

For each $1 \leq k \leq K$, we partition the set $\{0,1\}^K$ by clustering together indices $\widetilde{h}$ that have identical columns in $\widetilde{M}_{j_k}$. Denote this partition as $\mathcal{P}_k$. Also, define $\mathcal{Q}_k$ as the intersection of all partitions $\mathcal{P}_l$ for $l \leq k$:

$$\mathcal{Q}_k := \cap_{l \leq k} \mathcal{P}_l = \{\cap_{l \leq k} P_l : P_l \in \mathcal{P}_l, \quad \forall l \leq k\}.$$

Our main ingredient is to show the following Lemma, which uniquely determines the label permutation $\tau : [K] \rightarrow [K]$ from the partition $\mathcal{Q}_k$. We postpone its proof to Appendix D.

**Lemma 4.** *We claim the following: For any $k \leq K$,*

$$\mathcal{Q}_k = \{\{\sigma_\Pi(h_{1:k}, h') : h' \in \{0,1\}^{K-k}\}, \quad \forall h_{1:k} \in \{0,1\}^k\} \tag{10}$$

$$= \{\{\widetilde{h} : \widetilde{h}_{\tau(1)} = c_1, \ldots, \widetilde{h}_{\tau(k)} = c_k\}, \quad \forall (c_1, \ldots, c_k) \in \{0,1\}^k\} \tag{11}$$

*for some label permutation $\tau : [K] \to [K]$. Note that* (10) *implies $|\mathcal{Q}_k| = 2^k$, and $|P| = 2^{K-k}$ for any $P \in \mathcal{Q}_k$.*

Now, given the label permutation $\tau$, we identify each row of $\widetilde{\Gamma}$. Fix any $j \in [J]$. We use backwards induction on $k$ to show that $\widetilde{\gamma}_{j,\tau(k)} = \gamma_{j,k}$ for all $k$.

- We start with $k = K$. For any $P \in \mathcal{Q}_{K-1}$, Lemma 4 implies

$$P = \{\sigma_\Pi(h_{1:K-1}, 0), \sigma_\Pi(h_{1:K-1}, 1)\} = \{\widetilde{h} : \widetilde{h}_{\tau(1)} = c_1, \ldots, \widetilde{h}_{\tau(K-1)} = c_{K-1}\}$$

  for some $h_{1:K-1}, c_{1:K-1}$. Note that $\gamma_{j,k} = 1$ if and only if the columns of $M_j$ indexed by $\sigma_\Pi^{(-1)}(P) = \{(h_{1:K-1}, 0), (h_{1:K-1}, 1)\}$ are distinct. By the above characterization, this is equivalent to the columns of $M_j$ indexed by $P$ being distinct, which is equivalent to $\widetilde{\gamma}_{j,\tau(K)} = 1$. Hence, $\gamma_{j,K} = \widetilde{\gamma}_{j,\tau(K)}$.

- Now, for $k < K$, suppose that we know $\gamma_{j,l} = \widetilde{\gamma}_{j,\tau(l)}$ for $l \geq k + 1$. We determine $\widetilde{\gamma}_{j,k}$ by focusing on any $P \in \mathcal{Q}_{k-1}$. Here, when $k = 1$, we write $\mathcal{Q}_0 = \{[2^K]\}$ for notational convenience. Similar to the base case with $k = K$, the number of distinct columns indexed by $\sigma_\Pi^{(-1)}(P)$ in $M_j$ are equal to $2^{\sum_{l=k}^K \gamma_{j,l}}$. This value must be equal to the number of distinct columns indexed by $P$ in $\widetilde{M}_j$. By the characterization of $P$ in (11), this value is equal to $2^{\sum_{l=k}^K \widetilde{\gamma}_{j,\tau(l)}}$. Hence,

$$\sum_{l=k}^K \gamma_{j,l} = \sum_{l=k}^K \widetilde{\gamma}_{j,\tau(l)},$$

  and we get $\gamma_{j,k} = \widetilde{\gamma}_{j,\tau(k)}$ by the induction hypothesis.

Thus, we have shown the $k$-th column of $\Gamma$ is identical to the $\tau(k)$-th column of $\widetilde{\Gamma}$.

*Step 3:* Identify $\mathbb{P}(H), \mathbb{P}_{j,h}$, and the latent DAG $\Lambda$.

Now, we identify the remaining model components by additionally using the subset condition. Without the loss of generality, assume $\tau$ is the identity permutation on $[K]$, which gives $\Gamma = \widetilde{\Gamma}$. Consequently, we omit the $\widetilde{\phantom{x}}$ notation for $\Gamma$ and $H$. Our main idea is to use the subset condition on $\Gamma$ to show that for each $k$, the set of columns indexed by $\widetilde{h}_k = 0/1$ must match that with $h_k = 0/1$:

$$\{\{\widetilde{h} : \widetilde{h}_k = 0\}, \{\widetilde{h} : \widetilde{h}_k = 1\}\} = \{\{\sigma_\Pi(h) : h_k = 0\}, \{\sigma_\Pi(h) : h_k = 1\}\}. \tag{12}$$

Without loss of generality, let $k = 1$. This is purely for notational convenience, and does not overlap with the indexing for the triangular $\Gamma$-matrix, which will be not used in this proof segment. Let $\mathcal{J}_1 := \{j \in [J] : \gamma_{j,1} = 1\}$ denote the row indices for the children of $H_1$. For each $j \in \mathcal{J}_1$, we cluster $\{0,1\}^K$ based on identical columns $\widetilde{M}_j$. Then, we merge the clusters by combining clusters that share common elements. By the subset condition on $\Gamma$, $(0, \widetilde{h}_{(-1)})$ belongs in the same merged cluster for any $\widetilde{h}_{(-1)} \in \{0,1\}^{K-1}$. To see this, observe that the subset condition implies that the column vector $\gamma_{\mathcal{J}_1,k}$ must include at least one zero, for all $k \neq 1$. Let $j'_k$ be a row index in $\mathcal{J}_1$ such that $\gamma_{j'_k,k} = 0$. Then, $X_{j'_k}$ is not a child of $H_k$, and the all-zero vector $0_K$ and the basis vector $e_k = (0, \ldots, 1, \ldots, 0)$ (with 1 in the $k$th coordinate) must belong in the same cluster. By repeating this argument for each $k > 1$, we can conclude that $0_K$ and $(0, \widetilde{h}_{(-1)})$ belong in the same cluster. Since for each $j \in \mathcal{J}_1$, $X_j$ measures the latent variable $H_1$, $0_K$ and $(1, \widetilde{h}_{(-1)})$ can never be in the same cluster. Thus, after merging, there are two final clusters: $\{\widetilde{h} : \widetilde{h}_1 = 0\}$ and $\{\widetilde{h} : \widetilde{h}_1 = 1\}$.

This procedure inputs the matrices $\widetilde{M}_j$ without any column indexing, and characterizes $\{\{\widetilde{h} : \widetilde{h}_k = 0\}, \{\widetilde{h} : \widetilde{h}_k = 1\}\}$ for all $k$. Recalling that $M_j = \widetilde{M}_j \Pi^{-1}$ have an identical set of columns as $\widetilde{M}_j$ and applying the same characterization to $M_j$, we get (12). Thus, the indexings $h$ and $\sigma_\Pi(h)$ are identical up to a potential swapping of the meaning of $H_k = 0/1$. Without loss of generality, ignore the sign-flipping and assume $\{\widetilde{h} : \widetilde{h}_k = 0\} = \{\sigma_\Pi(h) : h_k = 0\}$ and $\{\widetilde{h} : \widetilde{h}_k = 1\} = \{\sigma_\Pi(h) : h_k = 1\}$ for all $k$.

Now, given a matching column indexing, $\pi_h = \widetilde{\pi}_h$ follows from eq. (9). Having identified the latent proportion $\mathbb{P}(H)$, $\Lambda$ can also be identified up to Markov equivalence. This is the consequence of the faithfulness Assumption 2, and is a well-known property alongside many existing algorithms (Pearl, 2014; Spirtes et al., 2001). To this extent, it is possible to replace the faithfulness assumption to others that can recover the DAG $\Lambda$ from the pmf.

Finally, it remains to identify the conditional distributions $\mathbb{P}_{j,h}$ from the discretized values. As the above argument can be applied for any discretization $(C_{1,j}, \ldots, C_{\kappa_j,j})$, we have

$$\mathbb{P}_{j,h}(C_j) = \widetilde{\mathbb{P}}_{j,h}(C_j), \quad \forall C_j \in \mathcal{C}_j.$$

Note that all discretizations are subject to the same ambiguities (label permutation and sign flipping) since the alternative model $\widetilde{\mathbb{P}}$ does not depend on the specific discretization. This implies $\mathbb{P}_{j,h} = \widetilde{\mathbb{P}}_{j,h}$, since Lemma 2 states that $\mathcal{C}_j$ is a separating class that uniquely defines a probability measure on $\mathcal{X}_j$. Hence, all model components can be identified. $\square$

*Proof of Proposition 1.* Recall that $C_j$ denotes a baseline set such that $\mathbb{P}_{j,h}(C_j) > \mathbb{P}_{j,h'}(C_j)$ for any $h \succ h'$. Without the loss of generality, assume that $C_j \in \mathcal{C}_j = \{C_{1,j}, \ldots, C_{\kappa_j,j}\}$, and index it as $C_j = C_{1,j}$. Steps 1, 2 in the proof of Theorem 2 still hold, and it suffices to modify step 3. We proceed via induction, and show that for each $k \in [K]$, the column indices corresponding to $\{h : h_k = 0\}$ and $\{h : h_k = 1\}$ can be recovered from the matrix $M_{j_k}$ (without using the column indexing of $M_{j_k}$). Recall from step 2 of the proof of Theorem 2 that $\{j_1, \ldots, j_K\}$ denotes the row indices corresponding to the triangular structure $\Gamma_2$, and that $\mathcal{P}_k$ denotes the partition of the column indices $\{0,1\}^K$ based on identical columns of $M_{j_k}$.

First, for $k = 1$, the monotonicity conditions alongside $\gamma_{j_1} = (1, 0, \ldots, 0)$ imply

$$M_{j_1}\Big(1, (1, h_{(-1)})\Big) = \mathbb{P}_{j_1,(1,h_{(-1)})}(C_{j_1}) > \mathbb{P}_{j_1,(0,h_{(-1)})}(C_{j_1}) = M_{j_1}\Big(1, (0, h_{(-1)})\Big),$$

for any $h_{(-1)} \in \{0,1\}^{K-1}$. Then, $\{h : h_1 = 0\}$ corresponds to the column indices of $M_{j_1}$ with a smaller value of $M_{j_1}(1, \cdot)$.

Now, for any $2 \leq k \leq K - 1$, assume that the claim holds for $l \leq k$. We show that we can recover the set of indices $\{h : h_{k+1} = 0\}$ and $\{h : h_{k+1} = 1\}$ from $M_{j_{k+1}}$. By the induction hypothesis, we are able to map each partition $\mathcal{P}_k$ from step 2 of Theorem 2 to each $h_{1:k} \in \{0,1\}^k$. For any $h_{1:k} \in \{0,1\}^k$ and $s \in \{0,1\}^{K-k-1}$, the monotonicity assumption gives

$$M_{j_{k+1}}\Big(1, (h_{1:k}, 1, s)\Big) = \mathbb{P}_{j_{k+1},(h_{1:k},1,s)}(C_{j_{k+1}}) > \mathbb{P}_{j_{k+1},(h_{1:k},0,s)}(C_{j_{k+1}}) = M_{j_{k+1}}\Big(1, (h_{1:k}, 0, s)\Big).$$

Thus, for any $P \in \mathcal{Q}_k = \cap_{l \leq k} \mathcal{P}_l$ (that corresponds to $h_{1:k}$ in the sense of (10)), we can further partition it into two sets based on the value of $M_{j_{k+1}}(1, h)$ for $h \in \mathcal{P}$. The set with the larger probability must correspond to $(h_{1:k}, 1)$, and we denote it as $Q_{(h_{1:k}, 1)}$. Now, the set of column indices corresponding to $h_{k+1} = 1$ can be recovered as $\cup_{h_{1:k}} Q_{(h_{1:k}, 1)}$, and the induction is complete.

Given the column indexing, we again have $\pi_h = \mathbb{P}(H = h) = \widetilde{\mathbb{P}}(\widetilde{H} = h) = \widetilde{\pi}_h$ and the same argument in step 3 of Theorem 2 completes the proof. Note that while Theorem 2 has determined the column indexing up to sign-flipping, this proof does not suffer from sign-flipping. $\square$

### C.4 Proof of Theorems 3 and 4

We prove Theorem 3 under a generalized BLCM, where $H = (H_1, \ldots, H_k)$ are *categorical* latent variables (not necessarily binary). Our key observation is the following non-identifiability result for latent class models with two observed variables (taking general responses), which is motivated by Theorems 4.1, 4.2 in Hall & Zhou (2003).

**Lemma 5** (Non-identifiability of LCMs with $K = 1, J = 2$)**.** *Consider a latent class model (LCM) with one latent categorical variable $H_1 \in \Omega_1$ and two observed variables $X_1, X_2$ which are conditionally independent given $H_1$. Then, the model components $\theta := \big(\mathbb{P}(H_1), \mathbb{P}(X_1 \mid H_1), \mathbb{P}(X_2 \mid H_1)\big)$ are non-identifiable as there exists a continuum of alternative parameters that define an identical marginal likelihood.*

While Hall & Zhou (2003) consider binary latent variables and continuous observations, the argument extends to Lemma 5. We provide a formal constructive proof in Appendix D for completeness.

*Proof of Theorem 3.* We show that three observed children per latent variable is necessary for identifiability. It suffices to consider the case when there exists some latent variable $H_1$ that has exactly two observed children $X_1, X_2$. We prove non-identifiability by showing that there exists alternative model parameters that define the same pmf of $\mathbb{P}(X)$.

First, we claim that it suffices to construct alternative parameters that define the same $\mathbb{P}(X_1, X_2 \mid H_{(-1)})$. To see this, write the marginal likelihood as

$$\mathbb{P}(X) = \sum_{h=(h_1, h_{(-1)})} \mathbb{P}(H = h)\mathbb{P}(X_1, X_2 \mid H = h) \prod_{j \geq 3} \mathbb{P}(X_j \mid H_{(-1)} = h_{(-1)})$$

$$= \sum_{h_{(-1)}} \Big[ \sum_{h_1} \mathbb{P}(X_1, X_2 \mid H = h)\mathbb{P}(H_1 \mid H_{(-1)}) \Big] \mathbb{P}(H_{(-1)} = h_{(-1)}) \prod_{j \geq 3} \mathbb{P}(X_j \mid H_{(-1)} = h_{(-1)})$$

$$= \sum_{h_{(-1)}} \mathbb{P}(X_1, X_2 \mid H_{(-1)} = h_{(-1)})\mathbb{P}(H_{(-1)} = h_{(-1)}) \prod_{j \geq 3} \mathbb{P}(X_j \mid H_{(-1)} = h_{(-1)}).$$

The last equality follows by writing the summand inside the square bracket in the second row above as

$$\sum_{h_1 \in \Omega_1} \mathbb{P}(X_1, X_2 \mid H = h)\mathbb{P}(H_1 = h_1 \mid H_{(-1)} = h_{(-1)}) = \sum_{h_1 \in \Omega_1} \mathbb{P}(X_1, X_2, H_1 = h_1 \mid H_{(-1)} = h_{(-1)})$$

$$= \mathbb{P}(X_1, X_2 \mid H_{(-1)} = h_{(-1)}).$$

Thus, any alternative model with the same value of $\mathbb{P}(X_1, X_2 \mid H_{(-1)})$, $\mathbb{P}(H_{(-1)})$, and $\mathbb{P}(X_j \mid H_{(-1)})$ for $j \geq 3$ defines the same marginal distribution of $X$.

For each configuration $h_{(-1)} \in \prod_{k=2}^{K} \Omega_k$, Lemma 5 shows that we can further specify a distinct set of parameters

$$\widetilde{\mathbb{P}}(H_1 \mid H_{(-1)} = h_{(-1)}), \ \widetilde{\mathbb{P}}(X_1 \mid H_1, H_{(-1)} = h_{(-1)}), \ \widetilde{\mathbb{P}}(X_2 \mid H_1, H_{(-1)} = h_{(-1)})$$

that define the same value of $\mathbb{P}(X_1, X_2 \mid H_{(-1)} = h_{(-1)}) = \widetilde{\mathbb{P}}(X_1, X_2 \mid H_{(-1)} = h_{(-1)})$. Hence, there exists an alternative set of parameters, and the model is non-identifiable.

Note that the above argument immediately implies non-identifiability of both the graph $E = (\Gamma, \Lambda)$ as well as the conditional distributions $\mathbb{P}(X_1 \mid H), \mathbb{P}(X_2 \mid H)$. $\qquad\square$

*Proof of Theorem 4.* Suppose there exists some $k \neq l \in [K]$ such that $\gamma_k \succeq \gamma_l$. Without loss of generality, let $k = 1, l = 2$. Denoting the true latent pmf as $\pi$, one cannot distinguish the values $(\pi_{(0,0,h_{3:K})}, \pi_{(0,1,h_{3:K})})$ and $(\pi_{(1,0,h_{3:K})}, \pi_{(1,1,h_{3:K})})$ up to a common sign flip of $H_2$, for each realization $h_{3:K} \in \{0, 1\}^{K-2}$. To see this, for any $h_{3:K}$, define alternative proportion parameters

$$\widetilde{\pi}_{(0,0,h_{3:K})} := \pi_{(0,0,h_{3:K})}, \quad \widetilde{\pi}_{(0,1,h_{3:K})} := \pi_{(0,1,h_{3:K})},$$

$$\widetilde{\pi}_{(1,0,h_{3:K})} := \pi_{(1,1,h_{3:K})}, \quad \widetilde{\pi}_{(1,1,h_{3:K})} := \pi_{(1,0,h_{3:K})},$$

and alternative conditional distributions with

$$\widetilde{\mathbb{P}}_{j,(0,0,h_{3:K})} = \mathbb{P}_{j,(0,0,h_{3:K})}, \quad \widetilde{\mathbb{P}}_{j,(0,1,h_{3:K})} = \mathbb{P}_{j,(0,1,h_{3:K})},$$
$$\widetilde{\mathbb{P}}_{j,(1,0,h_{3:K})} = \mathbb{P}_{j,(1,1,h_{3:K})}, \quad \widetilde{\mathbb{P}}_{j,(1,1,h_{3:K})} = \mathbb{P}_{j,(1,0,h_{3:K})}, \quad \forall j \in [J].$$

The observed distribution under the alternative parameters is identical since

$$\mathbb{P}(X) = \sum_h \pi_h \prod_{j=1}^J \mathbb{P}_{j,h}(X_j) = \sum_h \widetilde{\pi}_h \prod_{j=1}^J \widetilde{\mathbb{P}}_{j,h}(X_j) = \widetilde{\mathbb{P}}(X).$$

Now, it suffices to check that the alternative conditional distributions $\widetilde{\mathbb{P}}_{j,\cdot}$ are faithful to the graphical structure in $\Gamma$. To see this, first consider any row-vector $\gamma_j$ with $\gamma_{j,2} = 0$. Then, $H_2$ is not measured and the conditional distributions under $\widetilde{\mathbb{P}}$ are identical to that under $\mathbb{P}$:

$$\widetilde{\mathbb{P}}_{j,(0,0,h_{3:K})} = \widetilde{\mathbb{P}}_{j,(0,1,h_{3:K})} = \mathbb{P}_{j,(0,\cdot,h_{3:K})},$$
$$\widetilde{\mathbb{P}}_{j,(1,0,h_{3:K})} = \widetilde{\mathbb{P}}_{j,(1,1,h_{3:K})} = \mathbb{P}_{j,(1,\cdot,h_{3:K})}.$$

Next, suppose that $\gamma_{j,2} = 1$. Then, by the assumption $\gamma_1 \succeq \gamma_2$, we also have $\gamma_{j,1} = 1$. Since the nondegeneracy Assumption 3 implies that the four conditional distributions in the above display are all distinct, the distribution $\widetilde{\mathbb{P}}_{j,\cdot}$ is faithful to the row-vector $\gamma_j$.

We note that the above construction generalizes the counterexample in Appendix C.3 of Kivva et al. (2021), which focuses on the specific instance of $K = 2, J = 2$. $\qquad\square$

## D   Proof of lemmas

*Proof of Lemma 1.* Recall the lower-triangular assumption on the first $K$ rows of $\Gamma$. Parametrize each conditional distribution by setting

$$\theta_{k,(h_1,\ldots,h_k)} := \mathbb{P}(X_k = 1 \mid H_{1:k} = (h_1,\ldots,h_k)) \in [0,1].$$

We proceed via an induction on the latent dimension $K$. The claim is trivial for $K = 1$. For any $K \geq 2$, suppose that the claim holds when there are $K - 1$ latent variables. Define a $2^K \times 2^K$ matrix $T$ as:

$$T(x,h) = \mathbb{P}(X \succeq x \mid H = h), \quad \forall x, h \in \{0,1\}^K.$$

Without the loss of generality, suppose that the row/columns of $T$ are indexed based on the increasing order of $\sum_{k=1}^K x_k 2^{k-1} / \sum_{k=1}^K h_k 2^{k-1}$ (from top to bottom/left to right). For example, when $K = 3$, the row/columns are indexed as follows:

$$(0,0,0) < (1,0,0) < (0,1,0) < (1,1,0) < (0,0,1) < (1,0,1) < (0,1,1) < (1,1,1).$$

Additionally, write

$$T = \begin{pmatrix} T_{00} & T_{01} \\ T_{10} & T_{11} \end{pmatrix},$$

where each $T_{ab}$ are $2^{K-1} \times 2^{K-1}$ matrices. See the below table for an example expression of $T$ when $K = 3$ (where each block corresponds to $T_{00}, T_{10}, T_{01}, T_{11}$):

For any $x_{1:K-1}$ and $h_{1:K-1}$, the lower-triangular structure implies

$$T_{00}(x_{1:K-1}, h_{1:K-1}) = \mathbb{P}(X_1 \succeq x_1, \ldots, X_{K-1} \succeq x_{K-1} \mid H_{1:K-1} = h_{1:K-1}, H_K = 0)$$
$$= \mathbb{P}(X_1 \succeq x_1, \ldots, X_{K-1} \succeq x_{K-1} \mid H_{1:K-1} = h_{1:K-1})$$

$$T_{00} = \quad \begin{array}{c|cccc} x \setminus h & (0,0,0) & (1,0,0) & (0,1,0) & (1,1,0) \end{array}$$

| $x \setminus h$ | $(0,0,0)$ | $(1,0,0)$ | $(0,1,0)$ | $(1,1,0)$ |
|---|---|---|---|---|
| $(0,0,0)$ | $1$ | $1$ | $1$ | $1$ |
| $(1,0,0)$ | $\theta_{1,0}$ | $\theta_{1,1}$ | $\theta_{1,0}$ | $\theta_{1,1}$ |
| $(0,1,0)$ | $\theta_{2,00}$ | $\theta_{2,10}$ | $\theta_{2,01}$ | $\theta_{2,11}$ |
| $(1,1,0)$ | $\theta_{1,0}\theta_{2,00}$ | $\theta_{1,1}\theta_{2,10}$ | $\theta_{1,0}\theta_{2,01}$ | $\theta_{1,1}\theta_{2,11}$ |
| $(0,0,1)$ | $\theta_{3,000}$ | $\theta_{3,100}$ | $\theta_{3,010}$ | $\theta_{3,110}$ |
| $(1,0,1)$ | $\theta_{1,0}\theta_{3,000}$ | $\theta_{1,1}\theta_{3,100}$ | $\theta_{1,0}\theta_{3,010}$ | $\theta_{1,1}\theta_{3,110}$ |
| $(0,1,1)$ | $\theta_{2,00}\theta_{3,000}$ | $\theta_{2,10}\theta_{3,100}$ | $\theta_{2,01}\theta_{3,010}$ | $\theta_{2,11}\theta_{3,110}$ |
| $(1,1,1)$ | $\theta_{1,0}\theta_{2,00}\theta_{3,000}$ | $\theta_{1,1}\theta_{2,10}\theta_{3,100}$ | $\theta_{1,0}\theta_{2,01}\theta_{3,010}$ | $\theta_{1,1}\theta_{2,11}\theta_{3,110}$ |

(Rows $(0,0,0)$–$(1,1,0)$ form $T_{00}$; rows $(0,0,1)$–$(1,1,1)$ form $T_{10}$.)

| $x \setminus h$ | $(0,0,1)$ | $(1,0,1)$ | $(0,1,1)$ | $(1,1,1)$ |
|---|---|---|---|---|
| $(0,0,0)$ | $1$ | $1$ | $1$ | $1$ |
| $(1,0,0)$ | $\theta_{1,0}$ | $\theta_{1,1}$ | $\theta_{1,0}$ | $\theta_{1,1}$ |
| $(0,1,0)$ | $\theta_{2,00}$ | $\theta_{2,10}$ | $\theta_{2,01}$ | $\theta_{2,11}$ |
| $(1,1,0)$ | $\theta_{1,0}\theta_{2,00}$ | $\theta_{1,1}\theta_{2,10}$ | $\theta_{1,0}\theta_{2,01}$ | $\theta_{1,1}\theta_{2,11}$ |
| $(0,0,1)$ | $\theta_{3,001}$ | $\theta_{3,101}$ | $\theta_{3,011}$ | $\theta_{3,111}$ |
| $(1,0,1)$ | $\theta_{1,0}\theta_{3,001}$ | $\theta_{1,1}\theta_{3,101}$ | $\theta_{1,0}\theta_{3,011}$ | $\theta_{1,1}\theta_{3,111}$ |
| $(0,1,1)$ | $\theta_{2,00}\theta_{3,001}$ | $\theta_{2,10}\theta_{3,101}$ | $\theta_{2,01}\theta_{3,011}$ | $\theta_{2,11}\theta_{3,111}$ |
| $(1,1,1)$ | $\theta_{1,0}\theta_{2,00}\theta_{3,001}$ | $\theta_{1,1}\theta_{2,10}\theta_{3,101}$ | $\theta_{1,0}\theta_{2,01}\theta_{3,011}$ | $\theta_{1,1}\theta_{2,11}\theta_{3,111}$ |

(Rows $(0,0,0)$–$(1,1,0)$ form $T_{01}$; rows $(0,0,1)$–$(1,1,1)$ form $T_{11}$.)

$$= \mathbb{P}\big(X_1 \succeq x_1, \ldots, X_{K-1} \succeq x_{K-1} \mid H_{1:K-1} = h_{1:K-1}, H_K = 1\big)$$
$$= T_{01}\big(x_{1:K-1}, h_{1:K-1}\big),$$

and we have $T_{00} = T_{01}$.

Now, note that $T$ can be obtained from the conditional probability matrix $\mathbb{P}(X_{1:K} \mid H)$ via elementary row operations. Also, by elementary column operations, $T$ can be reduced to

$$\bar{T} := \begin{pmatrix} T_{00} & 0 \\ T_{10} & T_{11} - T_{10}. \end{pmatrix}.$$

Hence, we have

$$\mathsf{rk}(\mathbb{P}(X_{1:K} \mid H)) = \mathsf{rk}(T) = \mathsf{rk}(\bar{T}),$$

and by the property of determinant for block matrices, it suffices to show that

$$\det(\bar{T}) = \det(T_{00}) \det(T_{11} - T_{10}) \neq 0.$$

For notational simplicity, for each $h_{(-1)} = h_{1:K-1} \in \{0,1\}^{K-1}$, define

$$\eta_{h_{(-1)}} := \theta_{K,(h_{(-1)},1)} - \theta_{K,(h_{(-1)},0)} \neq 0.$$

The last inequality is a consequence of Assumption 3. Noting that

$$T_{11}\big(x_{1:K-1}, h_{(-1)}\big) = \mathbb{P}\big(X_1 \succeq x_1, \ldots, X_{K-1} \succeq x_{K-1}, X_K = 1 \mid h_{(-1)} = h_{(-1)}, H_K = 1\big)$$
$$= T_{00}\big(x_{1:K-1}, h_{(-1)}\big)\theta_{K,(h_{(-1)},1)},$$
$$T_{10}\big(x_{1:K-1}, h_{(-1)}\big) = \mathbb{P}\big(X_1 \succeq x_1, \ldots, X_{K-1} \succeq x_{K-1}, X_K = 1 \mid h_{(-1)} = h_{(-1)}, H_K = 0\big)$$
$$= T_{00}\big(x_{1:K-1}, h_{(-1)}\big)\theta_{K,(h_{(-1)},0)},$$

we have

$$(T_{11} - T_{10})\big(x_{1:K-1}, h_{(-1)}\big) = T_{00}\big(x_{1:K-1}, h_{(-1)}\big)\eta_{(h_{(-1)})}.$$

In other words, each column of $T_{11} - T_{10}$ is equal to the corresponding column of $T_{00}$ times $\eta_{h_{(-1)}}$. For example, in our running example with $K = 3$, $T_{11} - T_{10}$ can be written as:

$$T_{11} - T_{10} = $$

| $x \setminus h$ | (0,0,1) | (1,0,1) | (0,1,1) | (1,1,1) |
|---|---|---|---|---|
| $(0,0,1)$ | $\eta_{00}$ | $\eta_{10}$ | $\eta_{01}$ | $\eta_{11}$ |
| $(1,0,1)$ | $\theta_{1,0}\eta_{00}$ | $\theta_{1,1}\eta_{10}$ | $\theta_{1,0}\eta_{01}$ | $\theta_{1,1}\eta_{11}$ |
| $(0,1,1)$ | $\theta_{2,00}\eta_{00}$ | $\theta_{2,10}\eta_{10}$ | $\theta_{2,01}\eta_{01}$ | $\theta_{2,11}\eta_{11}$ |
| $(1,1,1)$ | $\theta_{1,0}\theta_{2,00}\eta_{00}$ | $\theta_{1,1}\theta_{2,10}\eta_{10}$ | $\theta_{1,0}\theta_{2,01}\eta_{01}$ | $\theta_{1,1}\theta_{2,11}\eta_{11}$ |

Thus,

$$\det(T_{11} - T_{10}) = \det(T_{00}) \prod_{h_{(-1)} \in \{0,1\}^{K-1}} \eta_{h_{(-1)}},$$

and, we have

$$\det(\bar{T}) = \det(T_{00}) \det(T_{11} - T_{10}) = \det(T_{00})^2 \prod_{h_{(-1)} \in \{0,1\}^{K-1}} \eta_{h_{(-1)}} \neq 0.$$

Here, the inequality follows from the induction hypothesis and the observation that $\eta_{h_{(-1)}} \neq 0$. □

*Proof of Lemma 4.* We separately show (10) and (11). First, to show (10), note that the $h$-th column of $M_j$ (denoted as $M_j(:,h)$) is identical to the $\sigma_\Pi(h)$-th column of $\widetilde{M}_j$. Thus, each $P \in \mathcal{Q}_k$ can be represented as $P = \{\sigma_\Pi(h) : M_{j_k}(:,h) = M_{j_k}(:,h_0)\}$ for some $h_0$. Now, the simplification in (10) follows from recalling the form of $\gamma_{j,k}$ alongside part (b) of Assumption 3.

The second equality in (11) follows by induction on $k$. We inductively show that for each $k \leq K$, (11) holds.

- Suppose $k = 1$. As the corresponding row of $\Gamma_2$ is $\gamma_{j_1} = (1, 0, \dots, 0)$, we have $|\mathcal{Q}_1| = |\mathcal{P}_1| = 2$. Then, the nondegeneracy Assumption 3 gives $\sum_{l=1}^{K} \widetilde{\gamma}_{j_1,l} = 1$. In other words, $\widetilde{\gamma}_{j_1}$ is a standard basis vector $e_l$. Let $\tau(1)$ be this $l$. Then, (11) holds by the definition of $\mathcal{P}_1$.

- Now, for each $1 \leq k < K$, suppose that (11) holds. By the expression in (10), $\mathcal{Q}_{k+1} \not\subseteq \mathcal{Q}_k$. So, $\mathcal{P}_{k+1} \not\subseteq \mathcal{Q}_k$, and there must be some $l \neq \tau(1), \dots, \tau(k)$ such that $\widetilde{\gamma}_{j_{k+1},l} \neq 0$. Suppose that there exist $l_1, l_2 \neq \tau(1), \dots, \tau(k)$ with $\widetilde{\gamma}_{j_{k+1},l_1} = \widetilde{\gamma}_{j_{k+1},l_2} = 1$. This implies that each element in $\mathcal{P}_{k+1}$ must have identical values of $\widetilde{h}_{l_1}, \widetilde{h}_{l_2}$. But, recalling the induction hypothesis that any $P \in \mathcal{Q}_k$ must be written as $P = \{\widetilde{h} : \widetilde{h}_{\tau(1)} = c_1, \dots, \widetilde{h}_{\tau(k)} = c_k\}$ for some $c_{1:k}$, we have

$$2^{k+1} = |\mathcal{Q}_{k+1}| = |\mathcal{Q}_k \cap \mathcal{P}_{k+1}| \geq 2^k \times 2^2,$$

and hence contradiction. Thus, we must have exactly one $l \neq \tau(1), \dots, \tau(k)$ with $\widetilde{\gamma}_{j_{k+1},l} = 1$. Define $\tau(k+1)$ to be this $l$. Then, (11) follows since $\mathcal{Q}_{k+1} = \mathcal{Q}_k \cap \mathcal{P}_{k+1}$.

□

*Proof of Lemma 5.* First, note that it suffices to consider binary latent variables $H_1$. This is because, for the general categorical case with $|H_1| \geq 3$, we can fix $\mathbb{P}(H_1 = h), \mathbb{P}(X_j \mid H = h)$ to the correct values for $h \geq 3$ and use the non-identifiability of the binary case.

For notational convenience, write $p := \mathbb{P}(H_1 = 1) \in (0, 1)$. Following the convention in this paper, $(p, \mathbb{P})$ denotes the true model components, and $(\widetilde{p}, \widetilde{\mathbb{P}})$ denotes an alternative component that is constructed as follows. Let $\widetilde{p} \in (0, 1)$ and $\alpha_1, \alpha_2$ be any constants such that satisfy $\alpha_1 \alpha_2 = p(1-p)\frac{1-\widetilde{p}}{\widetilde{p}}$. Also, let $\beta_1, \beta_2$ be constants defined as

$$\beta_j = -\frac{\widetilde{p}}{1-\widetilde{p}}\alpha_j, \ j = 1, 2.$$

By construction, $\alpha_j$ and $\beta_j$ have different signs. For all $j = 1, 2$, define

$$\widetilde{\mathbb{P}}(X_j \mid H_j = 1) = \mathbb{P}(X_j) + \alpha_j \big( \mathbb{P}(X_j \mid H_j = 1) - \mathbb{P}(X_j \mid H_j = 0) \big),$$
$$\widetilde{\mathbb{P}}(X_j \mid H_j = 0) = \mathbb{P}(X_j) + \beta_j \big( \mathbb{P}(X_j \mid H_j = 1) - \mathbb{P}(X_j \mid H_j = 0) \big).$$

As $\mathbb{P}(X_j \mid H_j = 1) \neq \mathbb{P}(X_j \mid H_j = 0)$ by Assumption 3, this construction leads to a well-defined probability measure for infinitely many choices of $(\widetilde{p}, \alpha_1, \alpha_2)$. Note that taking $(\widetilde{p}, \alpha_1, \alpha_2) = (p, 1 - p, 1 - p)$ recovers the true parameters (i.e., $\widetilde{\mathbb{P}} = \mathbb{P}$), so we can take $(\widetilde{p}, \alpha_1, \alpha_2)$ that is close enough to the true parameters so that all probabilities are positive.

Now, it suffices to verify that these choices lead to an identical marginal likelihood, and we separately show

$$\mathbb{P}(X_j) = \widetilde{\mathbb{P}}(X_j), \quad \mathbb{P}(X_1, X_2) - \mathbb{P}(X_1)\mathbb{P}(X_2) = \widetilde{\mathbb{P}}(X_1, X_2) - \widetilde{\mathbb{P}}(X_1)\widetilde{\mathbb{P}}(X_2).$$

The first claim follows from plugging-in the definition of $\beta_j$ to get

$$\widetilde{\mathbb{P}}(X_j) = \mathbb{P}(X_j) + \big( \widetilde{p}\alpha_j + (1 - \widetilde{p})\beta_j \big) \big( \mathbb{P}(X_j \mid H_j = 1) - \mathbb{P}(X_j \mid H_j = 0) \big) = \mathbb{P}(X_j).$$

To show the second claim, we expand each sides in terms of $\mathbb{P}(X_j \mid H_j = 1), \mathbb{P}(X_j \mid H_j = 0)$ and write

$$\mathbb{P}(X_1, X_2) - \mathbb{P}(X_1)\mathbb{P}(X_2) = p(1 - p) \prod_{j=1}^{2} \big( \mathbb{P}(X_j \mid H_j = 1) - \mathbb{P}(X_j \mid H_j = 0) \big),$$

$$\widetilde{\mathbb{P}}(X_1, X_2) - \widetilde{\mathbb{P}}(X_1)\widetilde{\mathbb{P}}(X_2) = \widetilde{p}(1 - \widetilde{p}) \prod_{j=1}^{2} \big( \widetilde{\mathbb{P}}(X_j \mid H_j = 1) - \widetilde{\mathbb{P}}(X_j \mid H_j = 0) \big)$$

$$= \widetilde{p}(1 - \widetilde{p})(\alpha_1 - \beta_1)(\alpha_2 - \beta_2) \prod_{j=1}^{2} \big( \mathbb{P}(X_j \mid H_j = 1) - \mathbb{P}(X_j \mid H_j = 0) \big).$$

The claim follows noting that the two RHS above are identical by plugging in

$$\widetilde{p} = \frac{\widetilde{p}}{1 - \widetilde{p}} \frac{\alpha_1}{\alpha_1 - \beta_1}, \quad 1 - \widetilde{p} = \frac{\alpha_2}{\alpha_2 - \beta_2},$$

and using the identity $\alpha_1 \alpha_2 = p(1 - p) \frac{1 - \widetilde{p}}{\widetilde{p}}$.

$\square$

# E  Simulation details

## E.1  True parameters

The structural parameters are set as in the main text and here we describe the true distributional assumptions. The latent proportions $\pi$ under each latent DAG $\Lambda$ are set as follows:

(a) Chain: $\mathbb{P}(H_1 = 1) = 2/3$, $\mathbb{P}(H_2 = H_1 \mid H_1) = 2/3$ for both $H_1 = 0, 1$, $\mathbb{P}(H_3 = H_2 \mid H_2) = 2/3$ for both $H_2 = 0, 1$.

(b) Collider: $\mathbb{P}(H_1 = 1) = \mathbb{P}(H_3 = 1) = 2/3$,

$$\mathbb{P}(H_2 = 1 \mid H_1, H_3) = \begin{cases} 0.2 & \text{if} \quad (H_1, H_3) = (1, 1), \\ 0.4 & \text{if} \quad (H_1, H_3) = (1, 0), \\ 0.6 & \text{if} \quad (H_1, H_3) = (0, 1), \\ 0.8 & \text{if} \quad (H_1, H_3) = (0, 0). \end{cases}$$

(c) Dependent: consider an hierarchical binary latent variable $H_0$ such that $\mathbb{P}(H_0 = 1) = 2/3$, and define $\mathbb{P}(H_a = H_0 \mid H_0) = 2/3$ for all $a = 1, 2, 3$. Define $\pi = \mathbb{P}(H_1, H_2, H_3)$ by marginalizing out $H_0$.

The true conditional probabilities $\mathbb{P}_{j,h}$ are set as follows: $X_j \mid (H = h) \sim \text{Ber}(\mu_{j,h})$ for $j = 1, 2, 3, 4$, $X_j \mid (H = h) \sim \text{N}(\mu_{j,h}, 1)$ for $j = 5, 6$, $X_j \mid (H = h) \sim \text{Cauchy}(\mu_{j,h}, 1)$ for $j = 7, 8$, where $\mu_{j,h}$ is given in Table 7. Note that we use the same conditional probabilities $\mathbb{P}_{j,h}$ for all $\Lambda$.

| $j \setminus h$ | (0,0,0) | (0,0,1) | (0,1,0) | (0,1,1) | (1,0,0) | (1,0,1) | (1,1,0) | (1,1,1) |
|---|---|---|---|---|---|---|---|---|
| 1 | 0.1 | 0.1 | 0.1 | 0.1 | 0.9 | 0.9 | 0.9 | 0.9 |
| 2 | 0.05 | 0.05 | 0.4 | 0.4 | 0.7 | 0.7 | 0.95 | 0.95 |
| 3 | 0.05 | 0.7 | 0.05 | 0.7 | 0.4 | 0.95 | 0.4 | 0.95 |
| 4 | 0.985 | 0.857 | 0.714 | 0.571 | 0.429 | 0.285 | 0.143 | 0.014 |
| 5 | -2 | -0.5 | -2 | -0.5 | 2 | 0.5 | 2 | 0.5 |
| 6 | -1.5 | -1.5 | 1.5 | 1.5 | -1.5 | -1.5 | 1.5 | 1.5 |
| 7 | 0.5 | -0.5 | 2 | -2 | 0.5 | -0.5 | 2 | -2 |
| 8 | 0.5 | 0.5 | 0.5 | 0.5 | 0.5 | 0.5 | 0.5 | 0.5 |

Table 7: True parameter values of $\mu_{j,h}$ for each $j, h$.

For the ablation studies, we consider the following three possible structures for $\Gamma$ while fixing the latent DAG $\Lambda$ to be chain ($H_1 \to H_2 \to H_3$). Here, $\Gamma^{\text{DT}}$ is the bipartite graph in Figure 1 and satisfies the condition in Theorem 2. The bold entries in $\Gamma^{\text{dense}}, \Gamma^{\text{sparse}}$ indicate the changes made from $\Gamma^{\text{DT}}$. Note that $\Gamma^{\text{dense}}$ and $\Gamma^{\text{sparse}}$ does not satisfy the necessary condition in Theorems 4, 3, respectively.

$$
\Gamma^{\text{DT}} = \begin{pmatrix} 1 & 0 & 0 \\ 1 & 1 & 0 \\ 1 & 0 & 1 \\ 1 & 1 & 1 \\ 1 & 0 & 1 \\ 0 & 1 & 0 \\ 0 & 1 & 1 \\ 0 & 0 & 0 \end{pmatrix}, \quad
\Gamma^{\text{dense}} = \begin{pmatrix} 1 & 0 & 0 \\ 1 & 1 & \mathbf{1} \\ 1 & 0 & 1 \\ 1 & 1 & 1 \\ 1 & 0 & 1 \\ 0 & 1 & \mathbf{1} \\ 0 & 1 & 1 \\ 0 & 0 & 0 \end{pmatrix}, \quad
\Gamma^{\text{sparse}} = \begin{pmatrix} 1 & 0 & 0 \\ 1 & 1 & 0 \\ 1 & 0 & \mathbf{0} \\ 1 & 1 & 1 \\ 1 & 0 & \mathbf{0} \\ 0 & 1 & 0 \\ 0 & 1 & 1 \\ 0 & 0 & 0 \end{pmatrix}.
\tag{13}
$$

We also modified the parameter values accordingly from that in Table 7 for $\Gamma = \Gamma^{\text{dense}}, \Gamma^{\text{sparse}}$.

## E.2 Algorithm details

While our identifiability results (Theorem 1 and Theorem 2) naturally lead to an algorithm, its practical implementation is challenging. The main point of concern is computing the rank $2^K$ tensor CP decomposition in (4). In general, computing tensor decompositions are challenging nonconvex optimization problems. While there exists methods such as alternating least squares or Jennrich's algorithm (Kolda & Bader, 2009; Harshman et al., 1970), its finite sample accuracy was not satisfactory for our purpose. Moreover, the computational complexity suffers from potentially large values of the latent and observed dimensions $K, J$. Based on these concerns, we instead implement a simple score-based estimation procedure. Our algorithm is a two-step procedure, where we first maximize the penalized likelihood to estimate the bipartite graph $\Gamma$ and the latent proportion vector $\pi$. Then, we recover the latent causal graph $\Lambda$ from the estimated latent proportions $\hat{\pi}$. Here, the first step utilizes Ma et al. (2023), and the second step is motivated by a similar procedure in Zhang et al. (2026). See Algorithm 1 for the general pipeline.

We elaborate on each step.

*Step 1: Discretize.* For the continuous responses with $j \geq 5$, discretize each entry by setting $Y_{j,h} := 1$ if and only if $X_{j,h} > 0$. The threshold can be chosen differently, e.g. one may threshold at the sample average $\bar{X}_j$.

---

**Algorithm 1:** Full algorithm to learn the model components.

---

**Data:** Observed data $X^{(1)}, \ldots, X^{(N)}$

Set the truncation parameter $\epsilon = 0.125$ and the number of pseudo-samples $N_{\text{sample}} = 2000$

    1. Define the discretized data $Y^{(1)}, \ldots, Y^{(N)}$ by thresholding at 0.

    2. Apply the penalized EM algorithm (Algorithm 1 in Ma et al. (2023)) to estimate $\pi$ and $\Theta$.

    3. For each $j \in [J], k \in [K]$, estimate $\gamma_{j,k}$ by (14)

    4. Given $\widehat{\pi}$, we generate $N_{\text{sample}}$ pseudo-samples of $H$ and estimate $\Lambda$ by the GES algorithm.

**Output:** Estimated DAG $(\Gamma, \Lambda)$ and latent proportion $\pi$.

---

*Step 2: Penalized EM.* For each $j, h$, define the Bernoulli success probability as $\theta_{j,h} := \mathbb{P}(Y_{j,h} = 1) \in (0,1)$ and let $\Theta$ be the $J \times 2^K$ matrix with entries $\theta_{j,h}$. Denoting $\ell_N(\pi, \Theta; Y)$ as the discretized log-likelihood (see eq. (9) in Ma et al. (2023)), we maximize the following penalized log-likelihood

$$\ell_N(\pi, \Theta; Y) - \lambda \sum_{j=1}^{J} \sum_{h \neq h'} \min(|\theta_{j,h} - \theta_{j,h'}|, \tau)$$

via a penalized EM algorithm. The above penalty (referred to as the grouped truncated lasso penalty) induces sparsity in each row of $\Theta$.

We modify Algorithm 1 in Ma et al. (2023) to compute the above estimator. Note that the cited work considers an additional degeneracy in terms of the proportions $\pi_h$ and encourages their sparsity by adding another penalty. As we consider $\pi_h > 0$ for all $h$ (see Assumption 3), we ignore this redundant penalty by setting $\widetilde{\lambda}_1 = 0$. Following the choices in (pg. 196, Ma et al., 2023), we set the grid for other tuning parameters as $\widetilde{\lambda}_2 \in \{1, 10, 10^2, 10^3\}, \tau \in \{0.05, 0.1\}$. As the EM algorithm is known to suffer from convergence up to local optima, we initialize both $\theta$ and $\pi$ to be close enough to the truth. We initialize them as the linear combination of the true parameter and random noise (each with weight 0.7, 0.3). This initialization also fixes the label permutation and sign-flipping ambiguities mentioned in Section 2. We set the convergence criteria of the EM algorithm as either satisfying (a) the $l^2$ norm of the proportion estimates $\widehat{\pi}$ are smaller than 0.005, or (b) the algorithm reaches the 10th iteration.

*Step 3: Learn $\Gamma$.* Under Assumptions 2 and 3, $\gamma_{j,k} = 0$ if and only if $\theta_{j,(h_{(-k)},1)} = \theta_{j,(h_{(-k)},0)}$ for all $h_{(-k)} \in \{0,1\}^{K-1}$. Thus, we use the estimated $\widehat{\theta}_{j,h}$s to recover $\gamma$. To account for the finite-sample estimation error for each entry of $\widehat{\theta}$, we estimate $\gamma_{j,k}$ based on the median of all possible values of $\widehat{\theta}_{j,(h_{(-k)},1)} - \widehat{\theta}_{j,(h_{(-k)},0)}$ as follows:

$$\widehat{\gamma}_{j,k} := \begin{cases} 1 & \text{if} \quad \text{median}\big(\{|\widehat{\theta}_{j,(h_{(-k)},1)} - \widehat{\theta}_{j,(h_{(-k)},0)}| : h_{(-k)}\}\big) > \epsilon, \\ 0 & \text{otherwise} \end{cases} . \tag{14}$$

In practice, we choose the threshold $\epsilon = 0.125$ based on the observation that the maximum threshold in the penalty function in step 2 is $\tau = 0.1$ (and allowing a small margin of error). Empirically, this choice of $\epsilon$ leads to the best accuracy for $\Gamma$ compared to other thresholds.

*Step 4: Learn $\Lambda$.* To learn the latent causal DAG $\Lambda$ from $\widehat{\pi}$, we generate 2000 pseudo-samples from the multinomial distribution on $\{0,1\}^K$ with probability $\widehat{\pi}$ and apply the Greedy Equivalence Search (GES) algorithm (Chickering, 2002). This structure learning method has been recently proposed in Zhang et al. (2026), alongside consistency gurantees. We implemented this procedure via Python, using the package `ges`. We compute the final SHD metric between the estimated CPDAG and true $\Lambda$ using the `count_accuracy` function from Kivva et al. (2021)[3].

**Remark 9** (Potential methodological extensions)**.** *Note that we assume that $K = 3$ is known for the above estimation pipeline. In practice, the latent dimension $K$ can be chosen based on standard information criteria*

---

[3]https://github.com/30bohdan/latent-dag

*(such as BIC, EBIC) as well as domain knowledge and interpretability, and we conduct experiments to assess the accuracy of BIC in the following subsection. Additionally, one may modify the above algorithm by considering finer discretization for continuous responses, which may improve estimation accuracy. Finally, to better initialize the EM in step 2, one may view the BLCM as a mixture model and develop spectral initialization by utilizing the double-triangular structure.*

### E.3   Selecting the number of latent variables $K$

We additionally report the accuracy of selecting $K$ via BIC. We consider the same simulation setting as in Section 4 and additionally focus on the chain graph $\Lambda$. Here, we select $K$ from a grid $\{2, 3, 4\}$, where the upper bound of 4 is chosen based on the double triangular condition (as there are a total of $J = 8$ observed variables). To be fair for all values of $K$, for each simulated data, we work under 20 random initializations and select the smallest BIC. We consider (i) the usual double-triangular graph $\Gamma$ (see Figure 1) as well as an (ii) altered non-double triangular graph with $\gamma_{3,3} = 1$ and $\gamma_{j,3} = 0$ for $j \neq 3$ (this is non-double triangular as $H_3$ is measured only once).

The results in Table 8 illustrate that under the double-triangular $\Gamma$ (that satisfies Theorem 1), $K$ can be selected with reasonable accuracy. For $N = 5000$, there is a tendency of under-selecting, but this improves for the larger sample size $N = 10000$. In contrast, when $\Gamma$ is non-double-triangular, we see a drastic drop in the accuracy for selecting $K$. This illustrates non-identifiability when the double-triangular condition is violated.

| $K \setminus N$ | DT | | Non-DT | |
|---|---|---|---|---|
| | 5k | 10k | 5k | 10k |
| 2 | 55 | 38 | 99 | 99 |
| **3** | 45 | 62 | 1 | 1 |
| 4 | 0 | 0 | 0 | 0 |

Table 8: Distribution (%) of the selected $K$ under a double-triangular (first two columns) and non-double-triangular (last two columns) bipartite graph $\Gamma$. The correct value is $K = 3$, and 300 replications are conducted.

### E.4   Runtime

| $\Lambda \setminus N$ | 1k | 5k | 10k |
|---|---|---|---|
| Chain | 1.6 | 3.2 | 5.1 |
| Collider | 2.5 | 5.4 | 8.0 |
| Dependent | 2.3 | 4.1 | 6.0 |

Table 9: Average runtime (in seconds) for steps 1-3 in Algorithm 1.

We report the average runtime (over the 300 replications) in Table 9. As Step 4 above was implemented in a different language and did not vary across simulation settings, we only report the runtime of steps 1–3. The simulations were implemented in `MATLAB` on a personal laptop device (Lenovo ThinkPad X1).

### E.5   Uncertainty

We additionally report box plots to complement the average values reported in Table 3. Note that the SHD is integer-valued, so we chose to report the box plots as opposed to the standard deviation. As the quantiles and outliers are also integer-valued, the box plots look very similar for all three sample sizes that we consider. Thus, we only report the box plots for $N = 5000$. Note that $\Gamma$ is a bipartite graph between 8 observed variables and 3 latent variables, so the SHD value of 2 means that two edges are incorrect among the 24

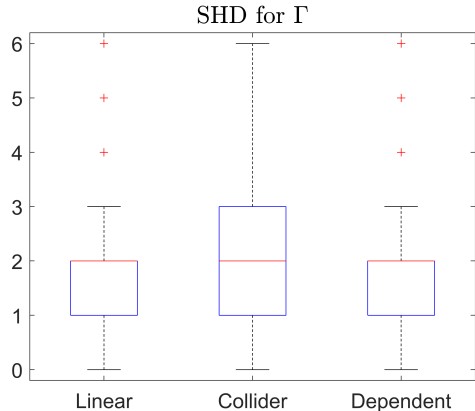 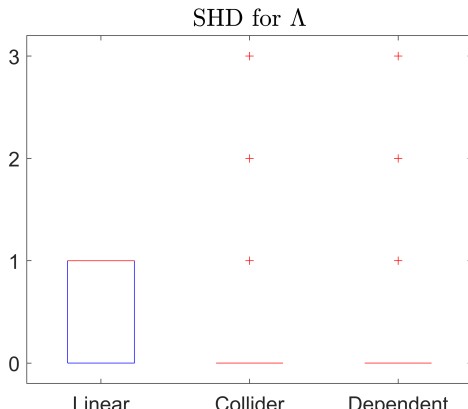

Figure 3: Box plots of the SHD values corresponding to Table 3, where $N = 5000$. **Smaller is better.**

possibilities. The SHD value for $\Lambda$ computes the discrepancy between the estimated CPDAG and $\Lambda$, and has the maximum value of 3.

## F   Real data details

The fraction subtraction dataset is publicly available through the R package `cdm` (George et al., 2016). We have focused on 17 of the 20 items provided in the dataset, following the example analysis in Chen et al. (2015).

Next, we provide the detailed estimates of the $17 \times 3$ bipartite graphical matrix $\Gamma$ under the fraction subtraction dataset. In Table 10, each row corresponds to one of the $J = 17$ questions, and each column corresponds to a latent skill. Our result resembles that in Table 11 of Chen et al. (2015) and we can understand each latent skill as: computing common denominator, writing integer as fraction, and subtracting integers. For example, questions 1-3 ask to compute fractions that do not involve integers $\frac{5}{3} - \frac{3}{4}, \frac{3}{4} - \frac{3}{8}, \frac{5}{6} - \frac{1}{9}$, so the estimated row of $\Gamma$ for these questions are $(1, 0, 0)$. However, for more complex questions such as question 17 which requires computing $3\frac{3}{8} - 2\frac{5}{6}$, all skills are required with the corresponding row in $\Gamma$ being $(1, 1, 1)$. Indeed, there exists only one pure child of the second skill in both estimates of $\Gamma$, as it is typically measured alongside the third skill.

## G   Justification for the binary latent variable assumption

The binary latent assumption simplifies our technical proof and algebraic requirements by allowing a clean, *non-generic* statement for Lemma 1. This is our key property that establishes the full-rankness of the augmented conditional distribution $\mathbb{P}(X_{1:K} \mid H_{1:K})$ when the corresponding observed-to-latent graph is lower-triangular. Its proof relies on showing a non-zero determinant $\det \mathbb{P}(X_{1:K} \mid H_{1:K}) \neq 0$. When we instead consider categorical latent variables that take $V > 2$ values, there are possible counterexamples.

For example, consider a toy setting with $V = 3, K = J = 2$ and a triangular structure with $H_1 \rightarrow X_1, H_2 \rightarrow X_1, X_2$, where each $X_k, H_k$ takes values in $\{0, 1, 2\}$. Setting $\theta_{x, h_1} := \mathbb{P}(X_1 = x \mid H_1 = h_1), \eta_{x, (h_1, h_2)} := \mathbb{P}(X_2 = x \mid H_1 = h_1, H_2 = h_2)$, we can show that $\det \mathbb{P}(X_{1:K} \mid H_{1:K}) \neq 0$ if and only if

$$\theta_{0,0}(\theta_{1,1} - \theta_{1,2}) + \theta_{0,1}(\theta_{1,2} - \theta_{1,0}) + \theta_{0,2}(\theta_{1,0} - \theta_{1,1}) \neq 0,$$

$$\eta_{0,(h,0)}(\eta_{1,(h,1)} - \eta_{1,(h,2)}) + \eta_{0,(h,1)}(\eta_{1,(h,2)} - \eta_{1,(h,0)}) + \eta_{0,(h,2)}(\eta_{1,(h,0)} - \eta_{1,(h,1)}) \neq 0, \quad \forall h = 0, 1, 2.$$

In general, we must assume that the values of such alternating polynomials (of degree $V - 1$) are nonzero for each observation $X_j$. While this is a generic statement that holds almost surely under fully categorical observations, such conclusions cannot be stated for the continuous or other general responses considered

| $j \setminus k$ | 1 | 2 | 3 |
|---|---|---|---|
| 1 | 1 | 0 | 0 |
| 2 | 1 | 0 | 0 |
| 3 | 1 | 0 | 0 |
| 4 | 0 | 1 | 0 |
| 5 | 1 | 0 | 1 |
| 6 | 0 | 0 | 1 |
| 7 | 0 | 0 | 1 |
| 8 | 1 | 0 | 0 |
| 9 | 1 | 0 | 1 |
| 10 | 1 | 0 | 1 |
| 11 | 0 | 1 | 1 |
| 12 | 0 | 1 | 1 |
| 13 | 0 | 1 | 1 |
| 14 | 1 | 1 | 1 |
| 15 | 0 | 1 | 1 |
| 16 | 1 | 1 | 1 |
| 17 | 1 | 1 | 1 |

Table 10: Estimated bipartite graph $\Gamma$ for the fraction subtraction dataset.

in our paper. Furthermore, this restricts the identifiability guarantees for parametric sub-models. For example, we cannot even take a linear parametrization $\theta_{x,h_1} = \mathbb{P}(X_1 = x \mid H_1 = h_1) = \beta_{0,x} + h_1 \beta_{1,x}$, under which the above equation for $\theta$ becomes exactly zero. In contrast, under binary latents with $V = 2$, this requirement is immediately satisfied by the usual nondegeneracy assumption for the conditional distributions (see Assumption 3(b)).

For Lemma 1 to hold under categorical latents, we also need each $X_k$ to take at least $V$ possible values. If $V > 2$, this rules out binary responses. This restriction can be troublesome in many practical scenarios for causal discovery in observational studies. For example, our real data example considers binary responses (correct/incorrect) arising in educational assessments, and it is known to be extremely challenging to establish identifiability when general categorical latent variables are used to model binary data. As another illustration, binary observations/confounders are also common in medical studies, e.g., the binary observation of a patient exhibiting a symptom or not is confounded by binary latent gene mutations.

