# OpenReview forum: "On Theoretical Identifiability of Binary Latent Causal Graphical Models"
_TMLR — Accepted by TMLR_

### Review · Reviewer_hjC5 · 2026-03-07

**Summary Of Contributions:**

The paper considers the problem of identifiability of causal graphical models with latent variables. Previous work on this either focussed on models with continuous variables with linear or additive constraints, or discrete variables with very stringent structural assumptions on the graph. The present work presents analysis in the discrete binary setting, where they prove identifiability under looser structural assumptions, identifying necessary and sufficient conditions on the causal graph structure for identifiability to hold.

__Strengths:__
- The paper is well-written, rigorously presented, and easy to follow for the most part.
- Empirical evidence for the theoretical result that double-triangular structure is key to sufficiency for identifiability, is provided in the "ablation experiment".

__Weaknesses:__
- The experiment section is weaker compared to the theoretical parts of the paper. From my understanding, the paper is not really a "methods" paper, so it is a little confusing what "our method" refers to in the experiments. This leaves me confused about the first and third experiment:
    - In the first experiment (__Experiments under latent structure__), my initial understanding of this experiment was to showing that provided $\Gamma$ (latent-to-observable relations) satisfies the double-triangular condition, then we can identify the causal graph regardless of the structure of $\Lambda$ (latent-to-latent relations). However, in Table 6 (Appendix E.3), the authors seem to make this into methodology comparison --- shouldn't the result indicate that as long as the structural assumptions of the graph are satisfied, we can identify the graph regardless of methodology?
    - Similarly in the third experiment (__Real data analysis__), the authors compare their "proposed" method with Chen et al. (2015), which, again is confusing as the paper is not a methods paper.
- Part of the claim of the paper is that it proves identifiability of causal graphical models under general structural assumptions, but the BLCM setting considered in this paper seems like quite a strict requirement. I am not sure how ubiquitous or realistic this setting is in real world data.

**Audience:**

Yes

**Audience Explanation:**

The paper makes progress in loosening the structural assumptions required for identification of causal graphical models. While the assumptions imposed are still far from real world settings, it identifies cases where one can prove such results rigorously, which I think will be of interest to some of TMLR's audience. I think overall, this paper is well-aligned with TMLR's objectives.

**Broader Impact Concerns:**

There are no negative ethical implications of the work as far as I can see.

**Claims And Evidence:**

Yes

**Claims Explanation:**

I believe the main strength of this paper is its mathematical rigor. The paper establishes the objects they are working with very clearly, with detailed assumptions and rigorous proofs. However, I think these results can be further supported by empirical evidence, which I don't think the paper does successfully, except for the second experiment (please see more details in __Requested Changes__).

**Requested Changes:**

My main criticism of the work is mainly regarding the experiments, and I think this can be improved by just focussing on validating the theoretical claims (like in the second "Ablation" experiment), rather than comparing or demonstrating methodology (the first and third experiments). Some suggestions for results that can be empirically demonstrated are, e.g.:
- If $\Gamma$ is double-triangular then $K$ is identifiable (Theorem 1). In contrast, one can maybe find some some $\Gamma$ that is not double-triangular, where $K$ can not be identified.
- If $\Gamma$ is double-triangular and $K$ is known (+ assumptions on $\Gamma_3$) then $\Gamma$ is identifiable (Theorem 2) -> already shown by experiment 2.
- BLCM is identifiable only if each latent variable has at least 3 observed children (Theorem 3). Possible way to demonstrate this is to have a BLCMs like in Figure 2 and empirically demonstrating that these are not identifiable.

I believe this will make the work more focussed and avoids narrative confusion.

---

> ### Author Response · Authors · 2026-04-20
> **Point-by-point response (part 1)**
>
> We thank the reviewer for carefully reading our manuscript and for your positive feedback. We appreciate that you found the paper well-written and rigorous, and that it well-aligns with the TMLR's objectives. We are also glad to hear that the paper was easy to follow.
>
> We also agree that making the experiments focus on validating the theory will help avoid narrative confusion. We reply to your comments point-by-point below.
>
> >The experiment section is weaker compared to the theoretical parts of the paper. From my understanding, the paper is not really a "methods" paper, so it is a little confusing what "our method" refers to in the experiments.
>
> Thank you for your question, and we apologize for the confusion. As you correctly pointed out, our paper is indeed a theoretical identifiability paper; while we do require a method for empirical evaluation, it is not our primary focus.
>
> >In the first experiment (Experiments under latent structure), my initial understanding of this experiment was to showing that provided $\Gamma$ (latent-to-observable relations) satisfies the double-triangular condition, then we can identify the causal graph regardless of the structure of $\Lambda$ (latent-to-latent relations). However, in Table 6 (Appendix E.3), the authors seem to make this into methodology comparison --- shouldn't the result indicate that as long as the structural assumptions of the graph are satisfied, we can identify the graph regardless of methodology?
>
> Regarding the first experiment (Experiments under varying latent structures), your understanding is correct, and its focus is to validate that we can accurately recover the causal graph regardless of the latent structure (i.e. **validate Theorem 2**). The auxiliary results in Appendix E.3 are **independent of the first experiment**, and were included to illustrate that the simulation setting is quite challenging by comparing the accuracy with the other baseline (mixture oracle). We agree that this comparison does not directly help the main narrative, and have removed the previous Appendix E.3.
>
> Regarding your question about identifying the graph regardless of the methodology, the mixture oracle results (see the previous Table 6) also indicate that the SHD error decreases as the sample size $N$ increases (although the errors are much larger than our implementation). So, loosely speaking, this does not contradict the theory.
>
> > Similarly in the third experiment (Real data analysis), the authors compare their "proposed" method with Chen et al. (2015), which, again is confusing as the paper is not a methods paper.
>
> We agree that the method comparison is unnecessary here and have removed the comparison with Chen et al. (2015). In other places of the manuscript, we have made minor word changes to remove the phrase “our method”.
>
>
> >Part of the claim of the paper is that it proves identifiability of causal graphical models under general structural assumptions, but the BLCM setting considered in this paper seems like quite a strict requirement. I am not sure how ubiquitous or realistic this setting is in real world data.
>
> We appreciate your concern regarding the restrictions on the BLCM setting. Mainly, this is a restriction that arises from establishing rigorous identifiability theory, and we have discussed this in Appendix G in the previous version. To elaborate, the BLCM setup simplifies our technical proof and algebraic requirements, especially leading to a clean statement of Lemma 1 (our key ingredient for sufficient conditions). While a *generic* analog of Lemma 1 can be established for more general categorical latent variable models, a clean non-generic statement is impossible due to technicalities.
>
> Furthermore, we would like to emphasize that the BLCM setup allows full generality of the responses. If we consider categorical latent variables with $V$ categories, typically one requires $|\mathcal{X}| \ge V$, which rules out binary observed variables. This can be a significant restriction in many observational applications such as education, including the fraction-subtraction dataset analyzed here.
>
> We agree that this is an important problem and have reiterated the practicality of the binary latent variable setup after it is formally introduced in Section 2 by adding the following real world analogy:
>
> *"positing binary latent variables are common for modeling discrete concepts, for example to model mastery/deficiency of cognitive skills in educational testing data."*

---

> > ### Author Response · Authors · 2026-04-20
> > **Point-by-point response (part 2)**
> >
> > > My main criticism of the work is mainly regarding the experiments, and I think this can be improved by just focussing on validating the theoretical claims (like in the second "Ablation" experiment), rather than comparing or demonstrating methodology (the first and third experiments). I believe this will make the work more focussed and avoids narrative confusion.
> >
> > Thanks for the excellent structural suggestion. We totally agree that connecting each experiment to each theorem will make the paper clearer, and we have included a brief organization at the beginning of the experiment section:
> >
> > *"Using simulated data, we validate the identifiability result in Theorem 2 and also validate the necessary conditions in Theorems 3, 4. Finally, we provide a real-data illustration of the double-triangular condition. Throughout the first two simulated experiments, we assume that the number of latents $K$ is given, and focus on evaluating the causal graph recovery. We conduct additional experiments regarding selecting $K$ in Section E.3, where we validate Theorem 1."*
> >
> > >If double-triangular then identifiable (Theorem 1). In contrast, one can maybe find some some that is not double-triangular, where can not be identified.
> >
> > This is an interesting suggestion. If we understand correctly, your question is to empirically show that $K$ may not be recovered when the double-triangular condition is violated. We have conducted additional experiments under (i) double-triangular and (ii) non-double-triangular graphs to estimate $K$ (see Appendix E.3). The results illustrate that the former (alongside a sufficiently large $N$) can select $K$ with reasonable accuracy where as it completely fails in the latter. This result validates Theorem 1.
> >
> > >If is double-triangular and is known (+ assumptions on ) then is identifiable (Theorem 2) -> already shown by experiment 2.
> >
> > Actually, our intention was to show this in experiment 1 (in the main text), as we clarified in response to your previous question (in part 1 of our response). To iterate, the purpose of experiment 1 is to verify Theorem 2 in the sense that we illustrate finite-sample accuracy under simulation settings that satisfy Theorem 2.
> >
> > >BLCM is identifiable only if each latent variable has at least 3 observed children (Theorem 3). Possible way to demonstrate this is to have a BLCMs like in Figure 2 and empirically demonstrating that these are not identifiable.
> >
> > Thank you for the suggestion. We actually already verified Theorem 3 (and also Theorem 4) in our current ablation study (second experiment). The intention of this second experiment was to assess violation of the double-triangular condition in Theorem 2. In particular, the sparse matrix $\Gamma^{\text{sparse}}$ is constructed so that it does not satisfy Theorem 3 as the latent variable $H_3$ has only 2 observed children. We have made this clearer in the text:
> >
> > *"In particular, we consider two alternate bipartite graphical structures where each necessary condition in Theorems 3 and 4 are violated."*

---

> > > ### Comment · Reviewer_hjC5 · 2026-04-26
> > >
> > > I thank the authors for their clarifications and realigning the experiments to highlight the theoretical contributions of the paper. The new experiments add to strengthening the theoretical results and overall, I believe the changes has made the paper's narrative clearer and stronger.

---

> > > > ### Author Response · Authors · 2026-04-26
> > > >
> > > > Thank you so much for promptly going through the revision. We agree that the changes make the paper's narrative clearer, thank you again for your helpful suggestions.

---

### Review · Reviewer_XaZk · 2026-03-18

**Summary Of Contributions:**

The paper presents a set of sufficient and (separately) necessary identifiability criteria for a specific class of causal graph. The class is a binary measurement model, characterised by a bipartite graph of binary latent variables connected to observables of arbitrary modality and possibly nonlinear dependence, allowing for any structure of directed causality between the latents. Motivation is given in an educational setting, for which the latents represent skill mastery and the observables are answers to questions. By exploiting the binary structure of latents and absence of hierarchical hidden structure, the authors are able to provide weaker criteria than “two pure children per latent” or assumptions on the latent causality.

Sufficient identifiability criteria are presented for the minimal number of latent nodes, latent-to-latent and latent-to-observable connections, and probability distributions P(latents) and P(observables|latents). These are based on the requirement of a double-triangular structure that relaxes the requirement of two pure children per latent, plus other requirements on the structure of the latent-to-observable graph including the subset condition. Necessary conditions show that all latents must have three children, and that the subset condition is necessary, which they link to the density and sparsity (respectively) of the latent-to-observable graph.

Two synthetic experiments are provided that demonstrate (i) okay (but not perfect) identifiability for a 3-latent 8-observable case with three different latent causal graphs, and (ii) degrading performance when the observable graph is modified to violate the necessary criteria in the directions of density and sparsity. Finally, the authors analyse an education example that satisfies their sufficient criteria (but not two-children-per-latent), in which the goal is to identify the latent-to-observable structure from real-world data, finding comparable results to a previous analysis.

**Additional Comments:**

This paper is very well written. I appreciate the flow structure and care taken to ensure that statements are precise, whilst providing well-considered remarks and examples to assist the readers’ intuition.

Section 4:
- When you calculate SHD, do you already account for Markov equivalence? This would be needed to ensure that it is possible to achieve SHD=0.
- In the appendix you describe the algorithm that requires you to start close to the true answer. If this is the case, and we are intentionally biasing the local minima in this way, it seems like it could confound the results. Is this the case?

**Audience:**

Yes

**Audience Explanation:**

Identifiability of causal structure is important for any setting that wishes to infer causal structure from data, which is particularly important for settings of interpretability and causally-faithful generalisation. The measurement model studied here is one that practitioners may wish to apply to practical problems, and would find these results informative for e.g. designing observables. Furthermore, there are several avenues for future work that readers may be interested in.

**Claims And Evidence:**

Yes

**Claims Explanation:**

The main results are the identifiability theorems. These are well formulated with precise statements, and the paper is well written and structured to ensure that they are easy for the reader to understand. The empirical evidence is reasonable although could be improved with better optimisation and diversity of examples, however in this case the purpose of experiments is only to validate the theoretical results, and so I consider them to be sufficient.

N.B. Whilst the Theorems seem reasonable, I haven't been able to validate their proofs in depth at this stage.

**Requested Changes:**

Essential:
- Theorem 3 assumes that the observed variables are categorical. Is this correct? If so, why is it needed? As you point out elsewhere, it seems that we can connect the continuous and categorical cases by imagining arbitrarily fine binning. Furthermore, I understood from the preceding paragraph that Theorem 3 applied to all categorical latents, not only BLCM. Since this is one of the four key results, and this is a stronger assumption than the other Theorems, please clarify these in the text.

Optional:
- It is notable that Theorem 1 only guarantees that there are no hidden graphs $\Lambda$ with fewer nodes $\tilde K < K$. However, absent of further assumptions, there could still be graphs with $\tilde K > K$. Theorem 2 then proceeds under the assumption that $K$ is known exactly. The paper already recognises this and briefly discusses why such a mismatch is difficult to avoid, and some situations in which $K$ is exactly identifiable (most notably of course if we assume that all graphs are double triangular, as we can just re-use the same rank analysis). However, I would like this mismatch to be emphasised a little more, as a casual reader might not understand that a model that meets the assumptions of Theorems 1 and 2 does not yet guarantee that it is identifiable, except in some minimal-latents sense.
- The title is a little misleading because we don’t consider general categorical latents, only binary ones, except possibly in Theorem 3. It’s not a major problem and I understand that it might not be possible to change, but if it is possible then I would prefer it be made more clear.
- I did not follow the argument for example 3 because it wasn’t obvious to me how we counted the number of equations and free parameters. Please can you clarify this?
- For the ablation studies in section 4, studying the marginal Hamming distance over all (identifiable and non-identifiable) entries somewhat obscures the effect being proposed. In principle we expect the signature that the performance drop is driven by the non-identifiable entries, and these are specifically the ones for which extra data does not help. For this relatively simple example, is it possible to categorise the entries as identifiable vs non-identifiable and test whether these things are true?
- Provide public code for experiments

---

> ### Author Response · Authors · 2026-04-20
> **Point-by-point response (part 1)**
>
> We thank the reviewer for carefully reading our manuscript and for your positive feedback. We appreciate that you found the paper well-written/structured, precise, and easy to read. We are also glad to hear that you enjoyed the overall flow, including the detailed remarks/examples. Please see our point-by-point responses to your comments below.
>
> >Theorem 3 assumes that the observed variables are categorical. Is this correct? If so, why is it needed? As you point out elsewhere, it seems that we can connect the continuous and categorical cases by imagining arbitrarily fine binning. Furthermore, I understood from the preceding paragraph that Theorem 3 applied to all categorical latents, not only BLCM. Since this is one of the four key results, and this is a stronger assumption than the other Theorems, please clarify these in the text.
>
> Thank you for the insightful question. Your understanding is correct that our previous Theorem 3 assumes categorical observed variables. This was because our previous proof was built upon a crucial lemma regarding non-identifiability of categorical-response latent class models with two measurements (Lemma 5 in Appendix C.4), which established local non-identifiability via a Jacobian argument that could not allow continuous/non-parametric observations.
>
> During the revision, we found a related key reference (Hall and Zhou, 2003, AoS) and realized that this result (Lemma 5) does not require categorical observations. Hence, by carefully adapting their result to our graphical model setting, we found that the previous categorical observed variable requirement is actually unnecessary, so Theorem 3 holds under the general responses considered in our paper. We sincerely appreciate the reviewer for asking this question, which motivated us to think deeper and establish this new result in (the modified) Theorem 3. We believe that this change will make the paper stronger and make this result on the necessary condition coherent with other parts of our paper.
>
> The updated theorem reads as follows:
>
> *"For a BLCM with known $K$ to be identifiable, it is necessary for each latent variable $H_k$ to have at least *three* observed children. This necessary condition also holds when the latent variables $H$ are categorical (i.e., polytomous and not binary)."*
>
> Here, we kept the statement regarding the categorical latents (as opposed to binary), as this is a more general statement and may be useful for future researchers. We have also modified the statement and proof of Lemma 5 accordingly, including constructing an explicit counterexample that serves as the proof.
>
>
> >It is notable that Theorem 1 only guarantees that there are no hidden graphs $\Lambda$ with fewer nodes $\tilde{K}<K$. However, absent of further assumptions, there could still be graphs with $\tilde{K}>K$. Theorem 2 then proceeds under the assumption that $K$ is known exactly. The paper already recognises this and briefly discusses why such a mismatch is difficult to avoid, and some situations in which $K$ is exactly identifiable. However, I would like this mismatch to be emphasised a little more, as a casual reader might not understand that a model that meets the assumptions of Theorems 1 and 2 does not yet guarantee that it is identifiable, except in some minimal-latents sense.
>
> Your understanding is correct, and we agree that re-emphasizing the point on identifying $K$ would be helpful for the casual reader. Now, on the bottom of page 5, we have spelled out the possibility that one can always create additional latent variables. We have also reminded the reader of this subtlety right after the proof of Theorem 1 (see the first sentence of Remark 4).
>
> >The title is a little misleading because we don’t consider general categorical latents, only binary ones, except possibly in Theorem 3. It’s not a major problem and I understand that it might not be possible to change, but if it is possible then I would prefer it be made more clear.
>
> Thanks for the suggestion. Our original intention for the title was to use the term "discrete" to denote both the binary latent variables and the causal graph. We agree that most of the paper considers binary latents, so we have changed the title to "On Theoretical Identifiability of Binary Latent Causal Graphical Models" to better reflect this and make the scope clear.

---

> > ### Author Response · Authors · 2026-04-20
> > **Point-by-point response (part 2)**
> >
> > >I did not follow the argument for example 3 because it wasn’t obvious to me how we counted the number of equations and free parameters. Please can you clarify this?
> >
> > Thank you for the question. In example 3, we count the number of equations and parameters associated to the structure $H_2 \to X_4, X_5$. Here, one parameter is required for $P(H_2)$, and four parameters are required for $P(X_j \mid H_2 = h), j = 4,5, h = 1,2$, (here we assume that the observations $X_4, X_5$ are also binary). This gives the total of 5 parameters. Regarding the number of equations, the pmf for $P(X_4, X_5)$ gives four equations (as $X_4, X_5$ are binary) and we remove one to account for the pmf requirement. We have spelled this out in the example to make this clear.
> >
> > >For the ablation studies in section 4, studying the marginal Hamming distance over all (identifiable and non-identifiable) entries somewhat obscures the effect being proposed. In principle we expect the signature that the performance drop is driven by the non-identifiable entries, and these are specifically the ones for which extra data does not help. For this relatively simple example, is it possible to categorise the entries as identifiable vs non-identifiable and test whether these things are true?
> >
> > Thank you for the insightful question. We have actually reported this exact result in Supplement E.5, but did not emphasize this enough in the main text. Indeed, our theoretical necessary conditions give insights into non-identifiable entries (arrows in the bipartite graph), and the result justifies this. We have now moved this table and the related discussion to the main text for better illustration. Please see Table 5 in the *ablation studies* paragraph in Section 4 (in the updated manuscript).
> >
> > >Provide public code for experiments
> >
> > Thanks for reminding us, we have now included the codes for the experiments in the .zip supplement.
> >
> > **Respone to minor questions in additional comments**
> >
> > >When you calculate SHD, do you already account for Markov equivalence? This would be needed to ensure that it is possible to achieve SHD=0.
> >
> > This is indeed true, and we do account for Markov equivalence to compute the SHD by following the usual definition that computes the distance between the equivalence class of PDAGs [1]. We made it clear by spelling this out in Section 5:
> >
> > *"The SHD computes the number of incorrectly estimated edges between the two graphs by comparing the distance between the equivalence classes of PDAGs [1]."*
> >
> > [1] Tsamardinos I, Brown LE and Aliferis CF (2006). The max-min hill-climbing Bayesian network structure learning algorithm. Machine Learning
> >
> > >In the appendix you describe the algorithm that requires you to start close to the true answer. If this is the case, and we are intentionally biasing the local minima in this way, it seems like it could confound the results. Is this the case?
> >
> > Thank you for the question. While we indeed worked with a warm initialization, this is for computational convenience to better find a global optimum, because the consistency of the global optimum is a natural consequence of our identifiability conclusion. We do would like to point out that the graph structures are not constrained to be the ground truth in our initialization. In general, fitting the EM algorithm under a random initialization suffers from local optima, and our preliminary experiments showed that it required re-running the algorithm many times, so we utilized the warm initialization for the reported experiments. However, we recognize that developing a good initialization strategy is also an important problem; we believe that it is possible to view the BLCM as a mixture model and develop a practical spectral initialization by utilizing the double-triangular structure, but this is beyond the scope of the current paper. We have included this remark at the end of Remark 9.

---

> > > ### Comment · Reviewer_XaZk · 2026-05-01
> > >
> > > Thank you for your responses. I agree that the modified theorem is valuable, bringing it into a coherent story with the others, and appreciate the clarifications, including in the title.
> > >
> > > I have an additional question on page 20 appendix C2 - the proof of Theorem 1 for arbitrary responses. You extend the proof from binary to arbitrary responses by imagining that we partition the space of Xj and assign an indicator function to whether Xj falls in the range Cj. You then apply nondegeneracy assumption 3. However, I can imagine cases in which assumption 3b is satisfied for a continuous variable but not any possible binary projection defined this way. For example, consider that Xj is a 1D real number and has three parents, giving 2^3=8 possible configurations for h=pa(Xj). Consider that each of these 8 configurations map onto a disjoint region of the real number line Xj. Then, it seems there is no way to define Cj such that assumption 3b is satisfied for the binary projection - there will always be multiple configurations of h that lead to the same distribution over Yj. Do you agree with this, and if so, how does it affect your result?

---

> > > > ### Author Response · Authors · 2026-05-01
> > > >
> > > > Thank you for carefully checking and appreciating the revision, including the modified theorem.
> > > >
> > > > To answer your additional question regarding the proof of Theorem 1, we believe that there may have been a misunderstanding. We first clarify that the set $C_j$ does not have to be a connected set. Regarding the example you provided, suppose that each of the 8 configurations are Uniform random variables where each $X_j \mid H=h$ is supported on $[0,1], [1,2], ..., [7,8]$. While no connected set $C_j$ can distinguish $P(X_j \in C_j \mid H=h)$, we can constructed a disconnected set, e.g. $C_j = \cup_{i=0}^7 [i, i+i/8]$. Then $P(X_j \in C_j \mid H=h)$ takes distinct values of $0, 1/8, ..., 7/8$ for each configuration of $h$.
> > > >
> > > > Actually, our intension of Assumption 3(b) was to directly assume $P(X_j \in C_j \mid pa(X_j) = h ) \neq P(X_j \in C_j \mid pa(X_j) = h')$ for every $h \neq h'$ and non-empty subset $C_j \subsetneq \mathcal{X}_j$ (i.e., for every non-trivial measurable subset $C_j$). Under this clarification, the proof of Theorem 1 should be straightforward. We apologize that this was not clear from our initial presentation, and we will clarify Assumption 3(b) in the final version.

---

> > > > > ### Comment · Reviewer_XaZk · 2026-05-06
> > > > >
> > > > > You are quite right, thank you for this clarification and quick response. Indeed, it seems reasonable that this assumption will commonly be satisfied for some disconnected set, and the proof then follows naturally. I am happy with this to be clarified in the main text.
> > > > >
> > > > > Minor textual comment on page 10 in the second line of example 4: should $\Lambda$ be the identity matrix instead of the empty set?

---

> ### Author Response · Authors · 2026-05-06
>
> Thank you for your quick reply, we are glad that you agree that the assumption is natural and does not cause an issue with the proof.
>
> Regarding your question about Example 4, $\Lambda = \phi$ (empty set) is actually correct because we consider no edges in the latent graph $\Lambda$ here. To clarify more on the notation, at the beginning of pg 4, we defined $\Lambda$ as the ''edges among the hidden variables $H$'' (as opposed to the adjacency matrix), so we believe the empty set symbol is natural to denote the latent graph with no edges. We greatly appreciate your careful reading and your insightful comments.

---

### Review · Reviewer_GfZK · 2026-04-08

**Summary Of Contributions:**

This paper studies strict identifiability of latent causal graphical models under a measurement-model assumption, with binary latent variables and general observed response types. The main contribution is a new double-triangular condition on the latent-to-observed graph $\Gamma$, under which the paper establishes identifiability of the latent dimension $K$, and, with additional conditions, identifiability of $\Gamma$, the latent graph $\Lambda$ (up to Markov equivalence), and the associated model distributions. The paper also provides necessary conditions, including that each latent variable must have at least three observed children and that $\Gamma$ must satisfy a subset/non-nesting condition.

**Strengths**
- The paper tackles an important and well-motivated problem in latent-variable causal discovery.
- The proposed double-triangular condition is a meaningful relaxation of the standard pure-child assumption, which gives the paper clear novelty on the theory side.
- The presentation is reasonably clear for a theory paper: the motivation is easy to follow, the running example is helpful, and the main results are organized well.

**Weaknesses**
- The scope is still limited by the measurement-model assumption and the binary-latent setup, which may restrict applicability outside the intended settings.
- The experimental section is useful, but mainly serves as a finite-sample illustration of the theory rather than a strong practical evaluation. In particular, the main simulation pipeline assumes $K$ is known.

**Additional Comments:**

I found the paper interesting and reasonably easy to follow overall. The problem is important, the main structural idea is useful, and the paper does a good job of positioning itself relative to prior work. My main suggestion is to better calibrate the practical interpretation of the current experiments and make the intended application scope even more explicit.

**Audience:**

Yes

**Audience Explanation:**

Yes. The paper addresses a central question in latent-variable causal discovery, namely when a latent causal model is strictly identifiable under weak structural assumptions. This is relevant to researchers working on causal discovery, latent variable models, and related areas in machine learning and statistics.

**Broader Impact Concerns:**

I do not have specific broader impact concerns.

**Claims And Evidence:**

Yes

**Claims Explanation:**

Overall, yes. The paper presents a clear theoretical contribution, states its assumptions explicitly, and supports the main message with both sufficient and necessary identifiability results. I also found the overall narrative clear: the paper explains well what is new relative to prior pure-child-based results and why the proposed graphical condition matters.

The empirical section is not the main strength of the paper, but it is consistent with the theoretical story. The simulations are averaged over 300 independent trials, the reported SHD improves with larger sample sizes when the proposed conditions hold, and the ablation study behaves as expected when those conditions are violated. The real-data example further helps illustrate that the double-triangular condition is practically relevant.

**Requested Changes:**

1. **critical**: Please make the empirical scope more explicit in the main text. The theory includes identifiability of $K$, but the main simulation pipeline assumes $K=3$ is known. The appendix notes this clearly and also mentions that in practice $K$ can be selected using information criteria such as BIC/EBIC or domain knowledge; this distinction should be stated more prominently in the main paper. The real-data section already follows this approach by fitting $K=2,3,4$ and choosing $K=3$ via BIC.

2. **would strengthen the paper**: Add a concise comparison table against the most relevant prior work. The paper already discusses these comparisons in text, but a compact table summarizing whether $K$ is known, whether pure children are required, whether arbitrary response types are allowed, and whether the result is strict or generic identifiability would make the novelty easier to assess.

3. **would strengthen the paper**: Expand the discussion of the practical scope of the assumptions. The assumptions are clearly stated and largely standard for the intended setting, but a more explicit discussion of when the measurement-model assumption is realistic, and when the binary-latent setup is especially appropriate, would help readers better understand the range of applications of the theory.

---

> ### Author Response · Authors · 2026-04-20
>
> We thank the reviewer for carefully reading our manuscript and for your positive feedback. We appreciate that you found the paper clear, the problem to be important, and the proposed condition to be novel. We are also glad to hear that you found the paper interesting and easy to follow. Please see our point-by-point responses to your comments below.
>
> >critical: Please make the empirical scope more explicit in the main text. The theory includes identifiability of $K$, but the main simulation pipeline assumes $K=3$ is known. The appendix notes this clearly and also mentions that in practice $K$ can be selected using information criteria such as BIC/EBIC or domain knowledge; this distinction should be stated more prominently in the main paper. The real-data section already follows this approach by fitting $K=2,3,4$ and choosing $K=3$ via BIC.
>
> Thank you for the suggestion, and we agree that it is important to state that the main simulations work under a known $K$. We have clarified in the main text that the scope of the experiments is on learning the graphical structures, given a correctly specified $K$.
>
> *"The primary purpose of the experiments is to better understand the consequences of identifiability results in simulated and real data… Throughout the first two simulated experiments, we assume that the number of latents $K$ is given, and focus on evaluating the causal graph recovery. We conduct additional experiments regarding selecting $K$ in Appendix E.3."*
>
> To address your concern that the main simulation pipeline assumes $K$ is known, we have also conducted additional experiments that focus on learning $K$ and reported it in the Appendix (see section E.3). To summarize, BIC results in a reasonable selection accuracy (62% when $N=10000$) of $K$ under the double-triangular structure.
>
>
> >Add a concise comparison table against the most relevant prior work. The paper already discusses these comparisons in text, but a compact table summarizing whether $K$ is known, whether pure children are required, whether arbitrary response types are allowed, and whether the result is strict or generic identifiability would make the novelty easier to assess.
>
> This is a great suggestion, and we agree that a comparison table would provide a clearer visualization of the assumptions and setup. We have synthesized the detailed literature review from Appendix A into a compact table focusing on the aspects you suggested and included it in the literature review section (please see the newly added Table 1 in Section 1.1; also attached below for convenience). We did not include a column on strict/generic identifiability, as the relevant prior work typically considered the notion of strict identifiability, and we felt that the column would be redundant.
>
> | Paper Criteria | Know K | Assume pure children | Response type restrictions |
> | :--- | :--- | :--- | :--- |
> | Kivva et al. (2021) | X | X | Continuous |
> | Chen et al. (2024) | X | O | Discrete |
> | Chen et al. (2025) | X | O | X |
> | Lee & Gu (2025) | Ο | $\Delta$ | X |
> | Ours | X | X | X |
>
> >Expand the discussion of the practical scope of the assumptions. The assumptions are clearly stated and largely standard for the intended setting, but a more explicit discussion of when the measurement-model assumption is realistic, and when the binary-latent setup is especially appropriate, would help readers better understand the range of applications of the theory.
>
> We appreciate this suggestion and agree that the additional discussion on the practicality of the assumptions would be helpful.
> The measurement model assumption is natural in domains such as psychometrics and medicine, where observable test responses of students (or symptoms of patients) do not directly cause one another, but rather are common effects of unobservable underlying skills (or diseases).
>
> Furthermore, as detailed in Appendix G, binary latent variables are not just a mathematical convenience that allows us to bypass restrictive full-rank or mixture-oracle assumptions; they are highly appropriate for modeling specific, prevalent scenarios like the mastery/deficiency of a cognitive skill or the presence/absence of a disease.
>
> We have added this discussion right after introducing the assumptions to better motivate the range of applications (see Section 2.1 on page 3).

---

> > ### Comment · Reviewer_GfZK · 2026-04-24
> >
> > Thank you for the detailed and constructive response. My main concerns have been addressed, and I appreciate the additional clarifications and experiments added in the revised manuscript. In particular, the revisions make the scope of the experiments clearer, improve the positioning relative to prior work, and strengthen the discussion of the assumptions and their practical relevance. Overall, I think the updated version is stronger and easier to read.
> >
> > One small remaining presentation issue is Table 1: the symbols X / O / △ are useful for compact comparison, but the meanings of X and O are not explicitly explained in the table caption or nearby text. Adding a short legend would make the table easier to interpret.

---

> > > ### Author Response · Authors · 2026-04-24
> > >
> > > Thank you so much for promptly going through the revision. We are glad to hear that your main concerns are addressed and you think the updated version is stronger/easier to read. Thank you again for your careful reading and helpful suggestions.
> > >
> > > Regarding Table 1, we will add the following legend to the caption in our final version.
> > >
> > > *"Legend: O = Yes/required, X = no/not-required, $\triangle$ = partial."*

---

### Author Response · Authors · 2026-04-20
**Summary of revision**

We sincerely thank the action editor and three reviewers for their evaluation of the manuscript. We are glad to hear that all reviewers agreed that the manuscript was clear and precise, well-written, and interesting. We also thank reviewers GfZK and XaZk for appreciating the importance of the problem and hjC5 for finding the paper well-aligned with TMLR.

Below, we summarize the main changes made in the revised manuscript.

1. Following the suggestion of Reviewer XaZk, we strengthened a theorem regarding necessary conditions for identifiability (Theorem 3). To elaborate, we removed the categorical response restriction, which provides a stronger conclusion (establishing a necessary condition under arbitrary responses) and makes the theorem coherent with other parts of the paper.

2. Following the feedback of Reviewers GfZK and hjC5, we included an additional simulation on selecting the number of latent variables, $K$. We illustrate both the (i) finite-sample recovery of $K$ when our double-triangular condition is satisfied, and the (ii) non-identifiability when the double-triangular condition is violated (please see Appendix E.3 in the revised manuscript).

3. Following the feedback of Reviewer hjC5, we have removed all methodological comparisons in the paper (in particular, we removed the previous Appendix E.3 on comparisons; this is replaced by the experiments on selecting $K$).

We have also incorporated all other comments raised by the reviewers, and we reply to each comment point-by-point in separate replies. All changes are **highlighted in blue** in the .pdf attachment.

---

### Decision · Action_Editor_Ri8j · 2026-05-16

**Recommendation:** Accept as is

**Additional Comments:**

This paper explores the challenge of identifiability in binary latent causal graphical models and introduces a novel double-triangular condition, relaxing standard assumptions such as the need for multiple univariates per latent variable. Reviewers unanimously agreed that the study makes a significant and technically rigorous theoretical contribution, presenting clear results with a well-structured argument and appropriate placement in the literature. Although the empirical evaluation is primarily illustrative, and its applicability is somewhat limited due to the measurement model assumptions and the binary latent variable setting, these limitations are acceptable given the theoretical focus of the paper. The authors addressed the reviewers' concerns through revisions, including clarifying assumptions, strengthening theoretical results, and improving the experimental section. Overall, the paper meets TMLR's criteria and is recommended for acceptance and I thus recommend acceptance of this paper.

**Audience:**

Yes

**Audience Explanation:**

The finds in the paper are mostly interesting to researchers on causal learning.

**Claims And Evidence:**

Yes

**Claims Explanation:**

This paper explores the challenge of identifiability in binary latent causal graphical models and introduces a novel double-triangular condition, relaxing standard assumptions such as the need for multiple univariates per latent variable. Reviewers unanimously agreed that the study makes a significant and technically rigorous theoretical contribution, presenting clear results with a well-structured argument and appropriate placement in the literature.